# Interferon lambda 4 impacts the genetic diversity of hepatitis C virus

M Azim Ansari[1‡], Elihu Aranday-Cortes[2‡], Camilla LC Ip[1], Ana da Silva Filipe[2], Siu Hin Lau[2], Connor Bamford[2], David Bonsall[3], Amy Trebes[1], Paolo Piazza[1], Vattipally Sreenu[2], Vanessa M Cowton[2], STOP-HCV Consortium, Emma Hudson[3], Rory Bowden[1], Arvind H Patel[2], Graham R Foster[4], William L Irving[5], Kosh Agarwal[6], Emma C Thomson[2], Peter Simmonds[3], Paul Klenerman[3], Chris Holmes[7], Eleanor Barnes[3], Chris CA Spencer[1], John McLauchlan[2†], Vincent Pedergnana[1,8†*]

[1]Wellcome Centre Human Genetics, University of Oxford, Oxford, United Kingdom; [2]MRC-University of Glasgow Centre for Virus Research, Sir Michael Stoker Building, Glasgow, United Kingdom; [3]Nuffield Department of Medicine and the Oxford NIHR BRC, University of Oxford, Oxford, United Kingdom; [4]Blizard Institute, Queen Mary University, London, United Kingdom; [5]National Institute for Health Research (NIHR) Nottingham Biomedical Research Centre, Nottingham University Hospitals NHS Trust and University of Nottingham, Nottingham, United Kingdom; [6]Institute of Liver Studies, King's College Hospital, London, United Kingdom; [7]Department of Statistics, University of Oxford, Oxford, United Kingdom; [8]Laboratoire MIVEGEC (UMR CNRS 5290, IRD, UM), Montpellier, France

**\*For correspondence:**
vincent.pedergnana@cnrs.fr

[†]These authors also contributed equally to this work
[‡]These authors also contributed equally to this work

**Group author details:**
STOP-HCV Consortium See page 13

**Abstract** Hepatitis C virus (HCV) is a highly variable pathogen that frequently establishes chronic infection. This genetic variability is affected by the adaptive immune response but the contribution of other host factors is unclear. Here, we examined the role played by interferon lambda-4 (IFN-λ4) on HCV diversity; IFN-λ4 plays a crucial role in spontaneous clearance or establishment of chronicity following acute infection. We performed viral genome-wide association studies using human and viral data from 485 patients of white ancestry infected with HCV genotype 3a. We demonstrate that combinations of host genetic variants, which determine IFN-λ4 protein production and activity, influence amino acid variation across the viral polyprotein - not restricted to specific viral proteins or HLA restricted epitopes - and modulate viral load. We also observed an association with viral di-nucleotide proportions. These results support a direct role for IFN-λ4 in exerting selective pressure across the viral genome, possibly by a novel mechanism.
DOI: https://doi.org/10.7554/eLife.42463.001

## Introduction

Hepatitis C virus (HCV) infects an estimated 71 million people worldwide (*World Health Organization, 2017*) and can lead to severe liver disease in chronically infected patients. The virus is highly variable and has been classified into seven distinct genotypes, and further divided into 67 subtypes, based on nucleotide sequence diversity (*Simmonds, 2004*). The factors that have driven the evolutionary path of HCV are multifactorial but undoubtedly are also shaped by host genetics. Because of its major health burden, determining how both host and viral genetics contribute to the outcomes of infection is critical for a better understanding of HCV-mediated pathogenesis (*Ploss and Dubuisson, 2012*) and the immune response to viral infection.

Using a systematic genome-to-genome approach in a cohort of chronically infected patients, we recently reported associations between an intronic single nucleotide polymorphism (SNP) rs12979860 in the interferon lambda 4 (*IFNL4*) gene (CC vs. non-CC) and 11 amino acid polymorphisms on the HCV polyprotein (*Ansari et al., 2017*) at a 5% false discovery rate (FDR) (*Benjamini and Hochberg, 1995*). This observation was unexpected since *IFNL4* is a member of the type III interferon (IFN-λ) family that act as cytokines as part of the innate immune system and therefore lack apparent epitope specificity (*Bruening et al., 2017*). These associations between polymorphisms on the HCV polyprotein and host *IFNL4* SNP rs12979860 genotypes are further intriguing given that variants within the *IFNL4* locus (including SNP rs12979860) contribute to HCV clinical and biological outcomes, including spontaneous virus clearance, response to IFN-based treatment, viral load and liver disease progression (*Aoki et al., 2015*; *Ge et al., 2009*; *Noureddin et al., 2013*; *Patin et al., 2012*; *Rauch et al., 2010*; *Suppiah et al., 2009*; *Tanaka et al., 2009*; *Thomas et al., 2009*). It is possible that the associations between the outcomes of HCV infection and the *IFNL4* locus are directly linked to its impact on the viral genome and the encoded polyprotein.

The intronic *IFNL4* SNP rs12979860 is in high linkage disequilibrium with other SNPs that may be more biologically relevant, including the exonic dinucleotide variant rs368234815 ($r^2$ = 0.975 CEU population, 1000 Genomes dataset) in *IFNL4*. This variant [ΔG > TT] causes a frameshift, abrogating production of functional IFN-λ4 protein (*Prokunina-Olsson et al., 2013*). Moreover, several amino acid substitutions within IFN-λ4 have been shown to alter its antiviral activity (*Bamford et al., 2018*; *Terczyńska-Dyla et al., 2014*). In particular, a common amino acid substitution (coded by the SNP rs117648444 [G > A]) in the IFN-λ4 protein, which changes a proline residue at position 70 (P70) to a serine residue (S70), reduces its antiviral activity in vitro (*Terczyńska-Dyla et al., 2014*). Thus, the combination of alleles at rs368234815 and rs117648444 creates four potential haplotypes, two that do not produce IFN-λ4 protein (TT/G or TT/A; IFN-λ4-Null) and two that result in production of two IFN-λ4 protein variants (ΔG/G; IFN-λ4-P70 and ΔG/A; IFN-λ4-S70). Patients harbouring the impaired IFN-λ4-S70 variant display lower hepatic interferon-stimulated gene (ISG) expression levels, which is associated with increased viral clearance following acute infection and a better response to IFN-based therapy, compared to patients carrying the more active IFN-λ4-P70 variant (*Eslam et al., 2017*).

In this study, we report a large number of associations between HCV-encoded amino acids across the viral polyprotein and host *IFNL4* SNP rs12979860, with 42 significant associations at a 5% FDR, increasing to 76 viral sites at a 10% FDR. The associations are observed in both structural and non-structural viral proteins and no enrichment of association signals is observed in any of the viral proteins or HLA restricted epitope regions. We also find an association with viral nucleotide content and certain dinucleotide frequencies, such as UpA (uracil base followed by adenine base). Finally, we demonstrate that *IFNL4* haplotypes coding for IFN-λ4-S70 and IFN-λ4-P70 variants differ in terms of their impact on viral load and viral amino acid polymorphisms, in agreement with the reduced antiviral activity of IFN-λ4-S70. Together these observations suggest that IFN-λ4 is a driver of HCV sequence diversity and modulator of viral load.

## Results

### Host and virus genetic structures

To ensure that host and virus population structures had a minimal impact on our results, we used paired genome-wide human and viral genetic data in a homogenous group of 485 patients with self reported white ancestry, infected with HCV genotype 3a from two cohorts [BOSON (*Foster et al., 2015*) and Expanded Access Program (EAP) (*Foster et al., 2016*) cohorts, $N_{BOSON}$ = 411, $N_{EAP}$ = 74, see *Supplementary file 1* and Materials and methods for a description of the cohorts]. To control for both human and virus population structures, we performed principal component analysis (PCA) on each of the host and viral genetic data (Materials and methods). The host PCA defined a largely homogenous group corresponding to self-reported white ancestry (*Figure 1—figure supplement 1a*). The first and second viral principal components (PCs) explained around 3% and 2% of variance in HCV nucleotide diversity respectively (*Figure 1—figure supplement 1b*), indicating a homogenous group of isolates as observed by the long terminal and short internal branches of the phylogenetic tree (*Figure 1—figure supplement 2a*). The viral sequences from the two cohorts

were non-randomly distributed on the tree as one clade was underrepresented in the EAP cohort sequences; this clade corresponded to isolates in the BOSON cohort from outside the United Kingdom (treeBreaker Bayes factor = 249, see Materials and methods for an explanation on how to interpret Bayes factor and *Figure 1—figure supplement 3a*). This observation was not reflected in host *IFNL4* SNP rs12979860 genotypes, which were randomly distributed on the viral phylogenetic tree (treeBreaker Bayes factor = 1.1, *Figure 1—figure supplement 3b*). However, we did observe associations between the host *IFNL4* SNP rs12979860 and the fifth and seventh viral PCs (p=1.3×10$^{-15}$ and 7.2 × 10$^{-9}$, respectively), which were not directed by host-virus population co-structuring, suggesting that the *IFNL4* locus drives HCV nucleotide diversity (*Figure 1—figure supplement 2b–d* and Appendix 1).

## The *IFNL4* locus affects virus-encoded amino acids at specific sites across the HCV polyprotein

A major advantage of determining entire HCV genomic sequence data is the possibility to perform footprinting analysis at a genome-wide scale. The nucleotide and amino acid frequencies at polymorphic viral sites in the two cohorts were similar and no systematic differences were observed (*Figure 1—figure supplement 4*). We used logistic regression to test for association between *IFNL4* SNP rs12979860 genotypes (CC vs. non-CC) and virus-encoded amino acids, including the first two viral PCs and the first three host PCs as covariates to account for host-virus population co-structuring. Presence or absence of each viral amino acid was used as the response variable; 977 tests were performed at 471 viral sites. To test for possible confounders we separately added each of the cirrhosis status of patients, cohorts (BOSON vs. EAP), gender and age to the model as covariates. These covariates were not associated with any specific amino acids at a 10% FDR (data not shown).

At a 5% FDR, 42 of the viral sites tested were associated with *IFNL4* SNP rs12979860, increasing to 76 sites at a 10% FDR (*Figure 1* and *Supplementary file 2*). This represented 1.4% at a 5% FDR and 2.5% at a 10% FDR of all the viral amino acids in the HCV genotype 3a polyprotein (N = 3021), reflecting a large impact of the host *IFNL4* locus on the amino acids encoded at variable sites on the viral polyprotein. The most associated viral site was at position 2570 in the NS5B protein (p=1.32×10$^{-8}$, log(OR)=1.19), as previously reported (*Ansari et al., 2017*). Notably, 26 of the 76 sites (34%) associated with the *IFNL4* SNP rs12979860 at a 10% FDR lie within the HCV E2 glycoprotein (Appendix 1 and *Figure 1—figure supplement 5*). However, we did not observe significantly enhanced enrichment or depletion for association signals in any specific viral protein, or in previously reported HLA restricted epitope regions in HCV genotype 3a (*von Delft et al., 2016*) (*Supplementary file 3* and Materials and methods).

To ensure that host-virus population co-structuring or some other systematic bias was not confounding our results, we performed the same tests against the HCV amino acids for 500 host SNPs from across the human genome with a minor allele frequency (MAF) similar to *IFNL4* SNP rs12979860 MAF, further referred to as 'the 500 frequency-matched SNPs'. In effect we performed 500 viral GWASs, one for each of the 500 frequency-matched SNPs. Using a 5% FDR (calculated independently for each of the 500 viral GWASs), we observed no significant associations for 491 of the host SNPs tested against the HCV polyprotein. The remaining nine host SNPs were associated with one HCV amino acid each (*Supplementary file 4*). However, these associations are likely to be false positives as multiple testing corrections were performed for each viral GWAS independently. Additionally, the distribution of P-values for the association tests between HCV amino acids and the 500 frequency-matched SNPs followed the null distribution of no associations (*Figure 1—figure supplement 6a*), confirming that there was no systematic bias in our analysis. By comparison, the distribution of the P-values for the association tests between HCV amino acids and *IFNL4* SNP rs12979860 deviated from the null distribution of no associations. This observation and the large number of HCV amino acids significantly associated with *IFNL4* SNP rs12979860 genotypes highlighted that the broad impact of the *IFNL4* locus on HCV-encoded amino acids was authentic and not driven by host-virus population co-structuring.

We then explored nucleotide sequences at the codon level to distinguish the impact of the *IFNL4* SNP rs12979860 on viral nucleotides as distinct from its effect on viral amino acids. We tested for associations between host *IFNL4* SNP rs12979860 and HCV codon changes from the most common codon to synonymous and non-synonymous codons (Materials and methods). For each HCV codon site with at least 20 synonymous and 20 non-synonymous codons (N = 348), we performed a logistic

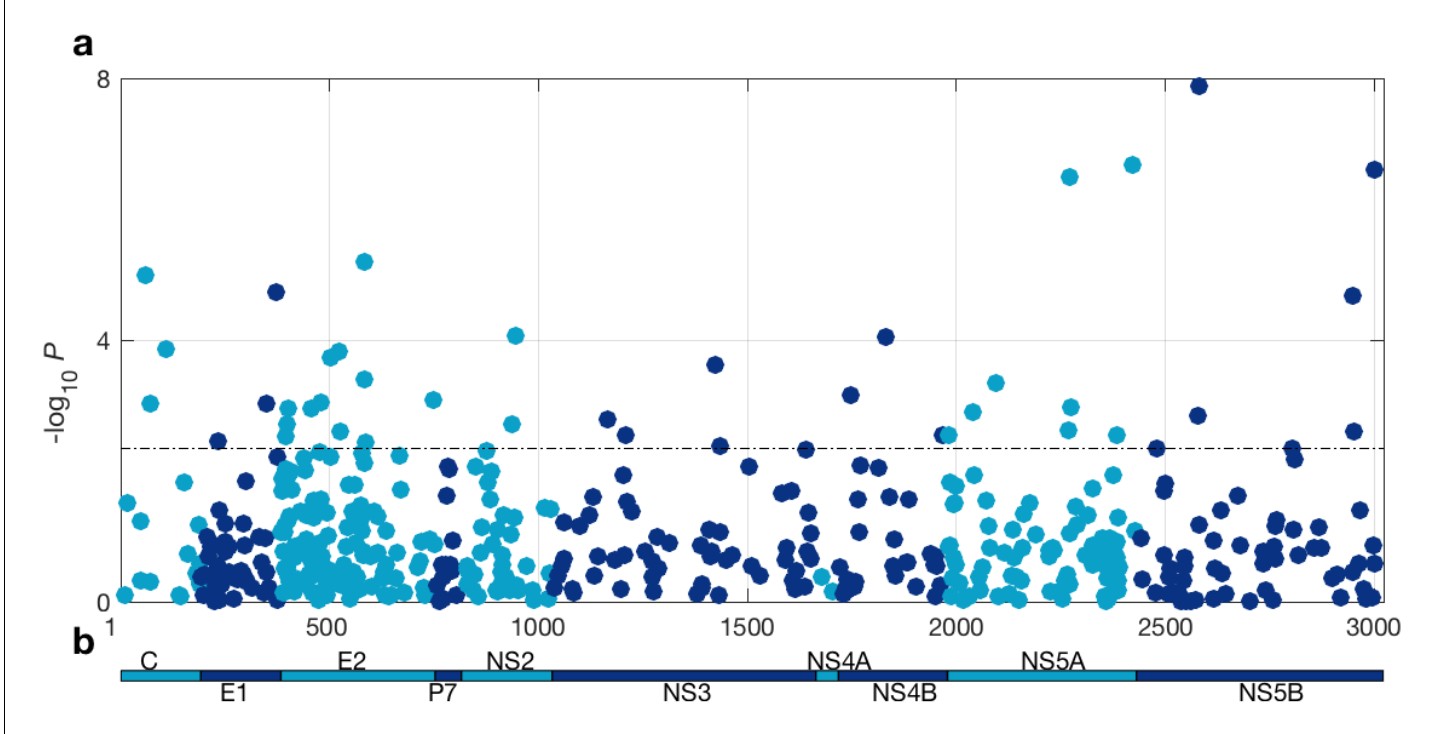

**Figure 1.** HCV genome-wide association study with *IFNL4* SNP rs12979860 genotypes (CC vs. non-CC). (a) Manhattan plot. The dashed line indicates 5% FDR. At this level 42 sites on the virus polyprotein are significantly associated with *IFNL4* SNP. (b) Schematic of the HCV polyprotein.
DOI: https://doi.org/10.7554/eLife.42463.002

The following figure supplements are available for figure 1:

**Figure supplement 1.** Host and viral principal components.
DOI: https://doi.org/10.7554/eLife.42463.003

**Figure supplement 2.** Association between viral PCs and *IFNL4* SNP rs12979860 genotypes in the combined cohort (N = 485).
DOI: https://doi.org/10.7554/eLife.42463.004

**Figure supplement 3.** Distribution of (a) cohort from which the sequences were obtained (BOSON N = 411 or EAP N = 74) and (b) host *IFNL4* SNP rs12979860 genotypes (CC or non-CC) on the virus phylogenetic tree.
DOI: https://doi.org/10.7554/eLife.42463.005

**Figure supplement 4.** Viral allele frequencies in the BOSON (N = 411) and EAP (N = 74) cohorts.
DOI: https://doi.org/10.7554/eLife.42463.006

**Figure supplement 5.** *IFNL4*-associated residues on the core E2 structure.
DOI: https://doi.org/10.7554/eLife.42463.007

**Figure supplement 6.** QQ-plots for association studies between viral amino acids and viral codons and host *IFNL4* SNP rs12979860 and 500 host SNPs chosen across the human genome with a minor allele frequency (MAF) similar to the *IFNL4* SNP rs12979860 MAF.
DOI: https://doi.org/10.7554/eLife.42463.008

**Figure supplement 7.** Effect size (beta) of *IFNL4* SNP rs12979860 genotypes (CC vs. non-CC) on the proportion of dinucleotide frequencies in the combined cohort (N = 485).
DOI: https://doi.org/10.7554/eLife.42463.009

**Figure supplement 8.** Effect size (beta) of *IFNL4* SNP rs12979860 genotypes (CC vs. non-CC) on the proportion of dinucleotide frequencies in the BOSON (N = 411) and EAP (N = 74) cohorts.
DOI: https://doi.org/10.7554/eLife.42463.010

regression including the first two viral PCs and the first three host PCs to test for association between *IFNL4* SNP rs12979860 (and the 500 frequency-matched SNPs as in the previous section) and changes from the most common codon to synonymous and non-synonymous codons (Materials and methods). We observed that non-synonymous changes at 16 viral codons were significantly associated with *IFNL4* SNP rs12979860 at a 5% FDR, increasing to 35 viral codons at a 10% FDR (*Supplementary file 5*). As expected the 500 frequency-matched SNPs did not show the same level of associations with HCV non-synonymous codon changes (*Figure 1—figure supplement 6b*).

We also observed that synonymous changes at two viral codons were significantly associated with *IFNL4* SNP rs12979860 at a 5% FDR, increasing to four viral codons at a 10% FDR (*Supplementary file 6* and *Figure 1—figure supplement 6c*). This indicates that the effect of the *IFNL4* locus on virus sequence diversity is mostly at the amino acid level, although a small impact on nucleotide substitutions cannot be excluded.

We hypothesised that the observed impact of *IFNL4* SNP rs12979860 on viral nucleotide sequences might be induced through dinucleotide sensing mechanisms. Most viruses suppress genomic CpG and UpA dinucleotide frequencies, supposedly to mimic host mRNA composition and avoid the immune response (*Simmonds et al., 2013*). To explore this possibility, we tested the association between the dinucleotide frequencies in each viral sequence and the host *IFNL4* SNP rs12979860 (Materials and methods). The viral UpA dinucleotide frequency (estimated as the ratio of observed to expected frequencies) was significantly lower in the host individuals with *IFNL4* SNP rs12979860 non-CC group compared to the CC group (p=$1.5\times10^{-6}$, *Figure 1—figure supplement 7*). By contrast, the viral UpG dinucleotide frequency was significantly higher in the *IFNL4* SNP rs12979860 non-CC group compared to the CC group (p=$1.5\times10^{-5}$). The viral CpC and CpA dinucleotide frequencies were also significantly different between the individuals with *IFNL4* SNP rs12979860 CC and non-CC genotypes (p=$3.3\times10^{-4}$ and p=$3.3\times10^{-4}$, respectively). Similar results were observed by analysing the cohorts independently (Appendix 1 and *Figure 1—figure supplement 8*).

## IFN-λ4 protein impacts on viral amino acid variation and viral load

We then investigated the impact of the different haplotypes of *IFNL4* on HCV amino acid diversity and viral load to refine its possible role. After imputing and phasing *IFNL4* rs368234815 and rs117648444 (Materials and methods), we observed three haplotypes: TT/G (IFN-λ4-Null); ΔG/G (IFN-λ4-P70) and ΔG/A (IFN-λ4-S70). HCV-infected patients were classified into three groups according to their predicted ability to produce IFN-λ4 protein: (i) no IFN-λ4 (two allelic copies of IFN-λ4-Null, $N_{BOSON}$ = 145, $N_{EAP}$ = 41), (ii) IFN-λ4–S70 (two copies of IFN-λ4-S70 or one copy of IFN-λ4-S70 and one copy of IFN-λ4-Null, $N_{BOSON}$ = 48, $N_{EAP}$ = 7), and (iii) IFN-λ4-P70 (at least one copy of IFN-λ4-P70, $N_{BOSON}$ = 218, $N_{EAP}$ = 26) (*Supplementary file 7*).

Since IFN-λ4-S70 can be distinguished phenotypically from IFN-λ4-P70 both in vivo and in vitro, we examined whether the *IFNL4* haplotypes had distinct effects on viral amino acid polymorphisms and viral load. We estimated the effect size of IFN-λ4-S70 and IFN-λ4-P70 relative to the IFN-λ4-Null haplotype on the presence and absence of the 76 amino acids associated with *IFNL4* SNP rs12979860 genotypes at a 10% FDR. We found that the estimated effect sizes of IFN-λ4-S70 were consistently smaller than those for IFN-λ4-P70 (*Figure 2*). Under the null hypothesis that there is no difference in the effect sizes of IFN-λ4-P70 and IFN-λ4-S70 variants on viral amino acid polymorphisms, we would expect the slope of the linear regression line (*Figure 2*) to have a value of one. However, the estimated slope of the best-fit line was significantly different from one (slope = 0.77, p=$9.6\times10^{-7}$, *Figure 2*). Additionally, we used bootstrapping to account for the uncertainty associated with the estimated effect sizes of IFN-λ4-P70 and IFN-λ4-S70 variants on HCV amino acid polymorphisms (Materials and methods). When estimating the slope of the line, we observed that the bootstrap 95% confidence interval [0.69, 0.99] for the slope of the line did not include one. Thus, we concluded that the impact of the host IFN-λ4-S70 variant on HCV-encoded amino acids was significantly smaller than the IFN-λ4-P70 variant.

We then investigated the effects of IFN-λ4 haplotypes on viral load. For this analysis, the EAP cohort was excluded as these patients had advanced liver disease with consistently lower viral loads relative to the BOSON cohort (*Figure 3—figure supplement 1*). We observed no difference in mean viral load between patients carrying IFN-λ4-S70 and IFN-λ4-Null haplotypes (p=0.61). However, the viral load in patients carrying IFN-λ4-P70 was significantly lower than in the other two groups ($P_{IFN-λ4-S70}$ = $1.6\times10^{-4}$ and $P_{IFN-λ4-Null}$ = $3.9\times10^{-10}$), with IFN-λ4-P70 conferring an approximately 2.3-fold decrease in viral load compared to IFN-λ4-S70 (mean for IFN-λ4-P70 = 2,905,333, IFN-λ4-S70 = 6,703,875 and IFN-λ4-Null = 6,256,523 IU/ml, *Figure 3a*).

We used a Bayesian approach to investigate the relationship between the effect sizes of the three IFN-λ4 haplotypes on viral load (*Figure 3b*). In essence, this method weighs up the evidence that the genetic effects of the IFN-λ4-Null, IFN-λ4-S70 and IFN-λ4-P70 haplotypes are the same or not relative to each other (Materials and methods). We tested five models; the effects of the three

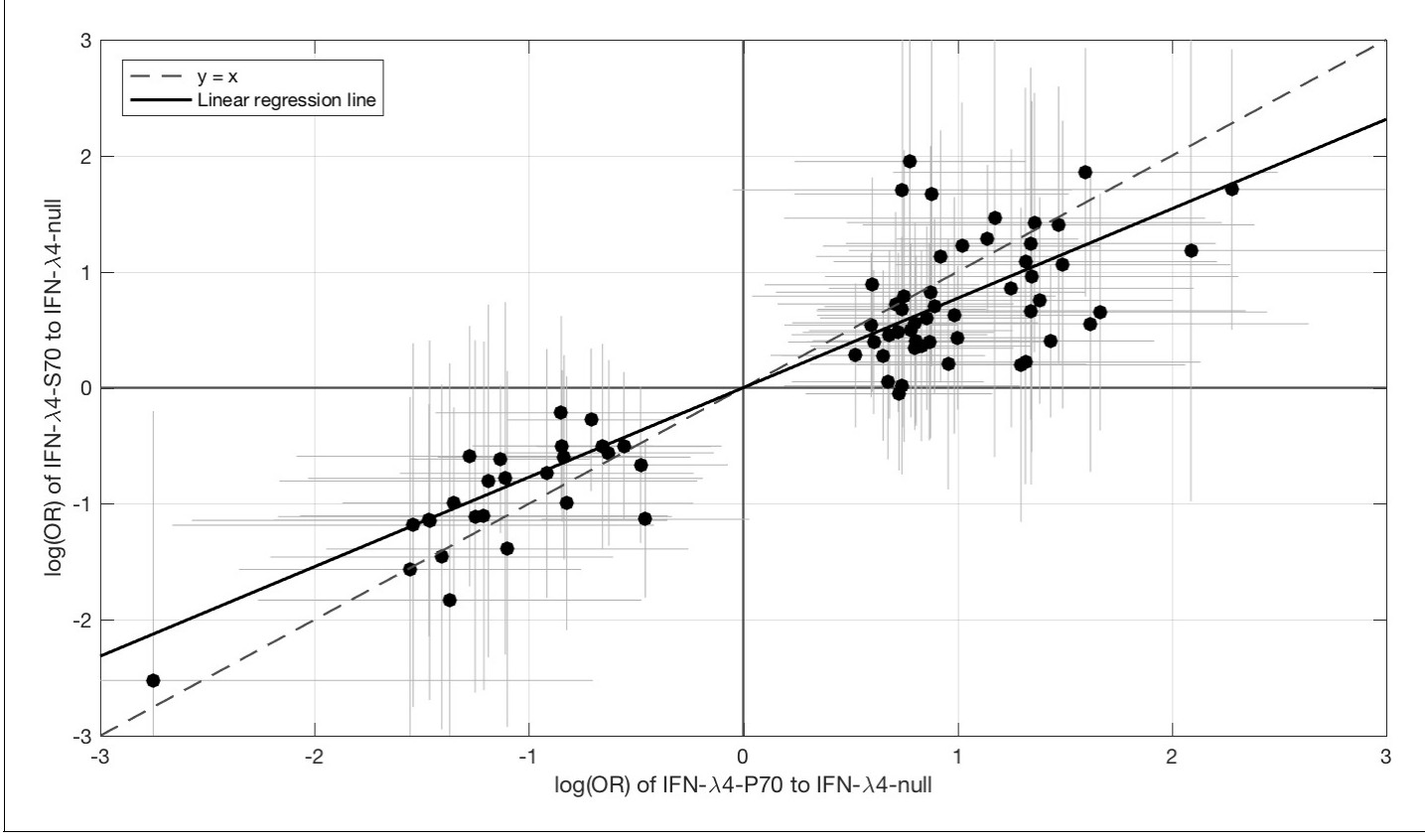

**Figure 2.** Comparison of the effect sizes (log(OR)) of host IFN-λ4 variants (IFN-λ4-S70 and IFN-λ4-P70 relative to IFN-λ4-Null) on HCV-encoded amino acids. The circles show the log(OR) estimates and the grey lines indicate the 95% confidence intervals. The dashed line is the y = x line which has a slope of one. The solid black line shows the linear regression line, which has a slope of 0.77 that is significantly different from one (y = x line, p=$9.6\times10^{-7}$).

DOI: https://doi.org/10.7554/eLife.42463.011

haplotypes are identical (model 1), the effects of IFN-λ4-P70 and IFN-λ4-Null are identical and different from the effect of IFN-λ4-S70 (model 2), the effects of IFN-λ4-S70 and IFN-λ4-Null are identical and different from the effect of IFN-λ4-P70 (model 3), all three haplotypes have different effect sizes (model 4) and the effects of IFN-λ4-P70 and IFN-λ4-S70 are the same but different from the effect of IFN-λ4-Null haplotype (model 5). Equal prior probabilities were used for all models. Model 3 had the highest posterior probability of 0.82 (*Figure 3b*).

We had previously reported an association between *IFNL4* SNP rs12979860 genotypes, the HCV-encoded amino acids at position 2414 in the polyprotein and viral load (*Ansari et al., 2017*). In the present study, we further stratified the analysis by *IFNL4* haplotypes. The viral serine (S) residue at site 2414 (S2414) was significantly enriched in patients coding for the IFN-λ4-P70 variant compared to IFN-λ4-S70 (p=$3.39\times10^{-03}$) and IFN-λ4-Null (p=$5.94\times10^{-09}$) coding patients. S2414 was present in 86% (211/244) of IFN-λ4-P70 carrying patients, 69% (38/55) of IFN-λ4-S70 carrying patients and 62% (114/185) of IFN-λ4-Null carrying patients. In HCV-infected patients with a serine residue at position 2414, we observed no significant difference in mean viral load between IFN-λ4-Null and IFN-λ4-S70 carriers (p=0.31, *Figure 3c*), but both groups had a significantly higher viral load than IFN-λ4-P70 carriers ($p_{IFN-\lambda4-Null}=2.7x10^{-9}$; $p_{IFN-\lambda4-S70}=1.6x10^{-10}$). However, no such association (p=0.49) was found in HCV-infected patients with a non-serine residue at this site (*Figure 3c*). We performed a Bayesian analysis that compared 58 possible models against each other (from all effect sizes being the same to all being different from each other). The model where only the 'IFN-λ4-P70 + S2414' group had an effect size different from the other groups (model 5) had the highest posterior probability of 0.33 (*Figure 3d* and Appendix1). Taken together, the combination of IFN-λ4-P70

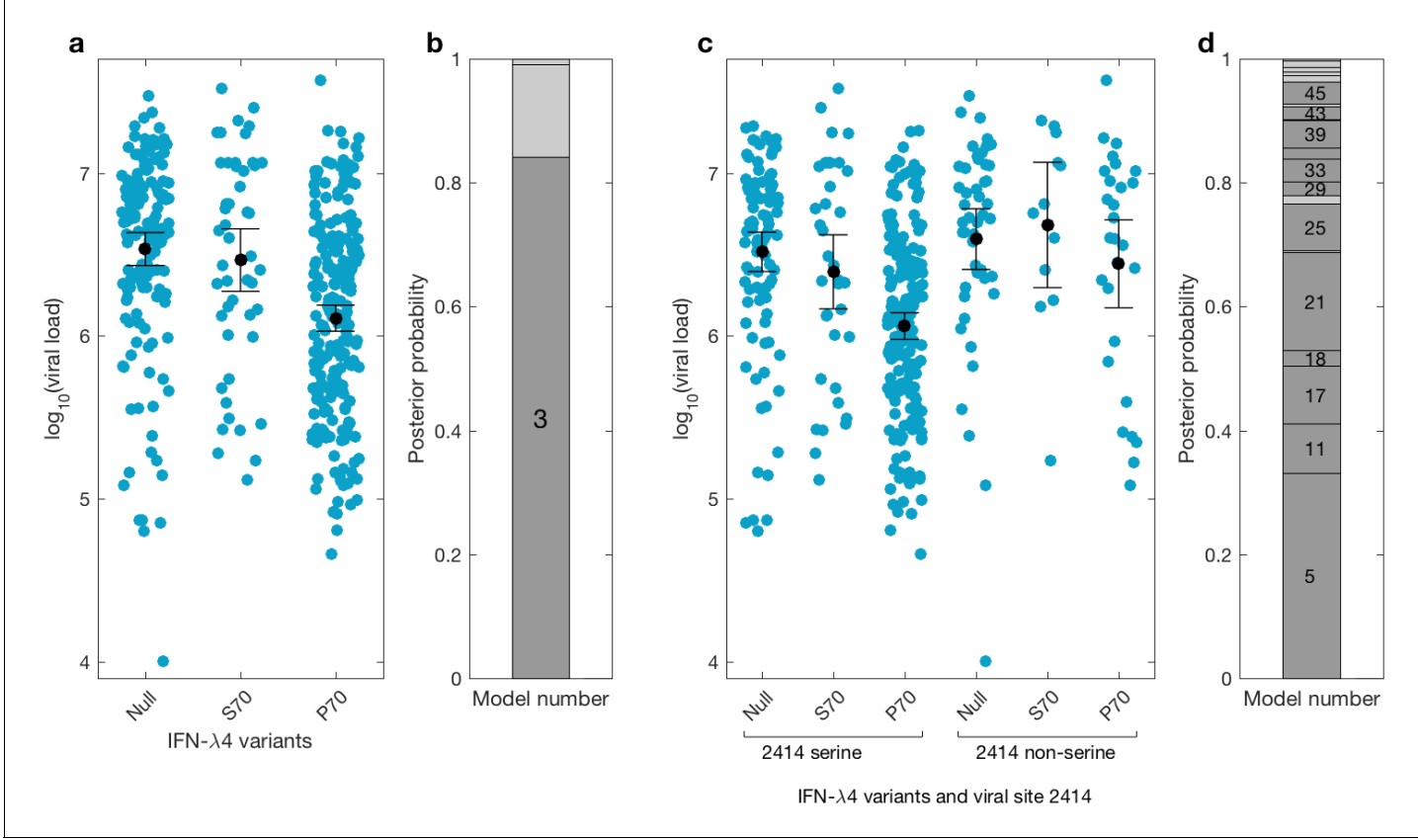

**Figure 3.** Bayesian model comparison of effect sizes of IFNL-λ4 variants on viral load in the BOSON cohort (N = 411). (a) Pretreatment viral load stratified by the host IFN-λ4 variants. The black dots and lines indicate the mean and 95% confidence interval (CI) for each group. (b) The posterior probability of the five tested models from (a) stacked on top of each other (from model 1 to model 5; posterior probabilities of models 1, 2 and 5 are too small to be labelled on this plot). Models where the posterior probability is higher or lower than the prior probability are coloured as dark grey and light grey respectively. Only model 3 (as indicated) has a posterior probability bigger than its prior probability and it assumes that the mean viral load is identical in IFN-λ4-Null and IFN-λ4-S70 groups and different from the mean viral load of IFN-λ4-P70 group. (c) Viral load stratified by the host IFNL-λ4 variants and the presence and absence of serine at the viral amino acid site 2414. The black dots and lines indicate the mean and 95% CI for each group. (d) The posterior probability of the 58 tested models from (c) stacked on top of each other (from model 1 to model 58; only models where the posterior probability is higher than the prior probability are labelled on this plot). Models where the posterior probability is higher or lower than the prior probability are coloured as dark grey and light grey respectively. Model 5 has the highest posterior probability and it assumes that the mean viral load is only different in 'IFN-λ4-P70 + 2414 serine' group and identical in other groups.

DOI: https://doi.org/10.7554/eLife.42463.012

The following figure supplement is available for figure 3:

**Figure supplement 1.** Pre-treatment viral load stratified by cohort.

DOI: https://doi.org/10.7554/eLife.42463.013

and S2414 conferred a 2.6-fold decrease in viral load compared to IFN-λ4-S70 and S2414 (mean viral load for IFN-λ4-P70 and S2414 = 2,376,747 IU/ml, and mean viral load IFN-λ4-S70 and S2414 = 6,093,167 IU/ml).

## Discussion

Here, we show that genetic variants in the human *IFNL4* locus drive widespread sequence changes across the entire HCV polyprotein - much larger than we previously reported (*Ansari et al., 2017*). We did not observe statistically significant enrichment of association signals in any specific HCV protein nor in HLA restricted epitope regions. This indicates that the host *IFNL4* locus, and hence the innate immune response, can influence the amino acid residues that are encoded at specific sites on the HCV polyprotein. The mechanism of action of IFN-λ4 in determining such selectivity is not

known. We also report an association of the *IFNL4* locus with synonymous codon variants, suggesting that this locus might also affect the HCV genome at the nucleotide sequence level. Finally, we report that the *IFNL4* haplotype coding for the IFN-λ4-S70 variant has a smaller impact on viral load and viral amino acid variation at associated sites compared to the haplotype coding for the more active IFN-λ4-P70 variant. This indicates that the *IFNL4* gene not only mediates HCV amino acid variation but also modulates viral load in patients. Hence, our findings extend the association between genetic variation in the *IFNL4* locus and outcome of HCV infection as well as hepatic disease (*Aoki et al., 2015*; *Ge et al., 2009*; *Noureddin et al., 2013*; *Patin et al., 2012*; *Rauch et al., 2010*; *Suppiah et al., 2009*; *Tanaka et al., 2009*; *Thomas et al., 2009*; *Eslam et al., 2017*).

We selected patients chronically infected with HCV genotype 3a and of self-reported white ancestry to limit the impact of human and viral population co-structuring in our analyses. We observed that 2.5% of HCV-encoded amino acids across the polyprotein were significantly associated with host *IFNL4* SNP at a 10% FDR. No such association was observed for 500 host SNPs chosen from across the human genome with a MAF similar to that for *IFNL4* SNP rs12979860. This indicated that the observed impact of *IFNL4* SNP rs12979860 on the viral sequences was not due to population structure or other systematic bias. Compared to our previous report (*Ansari et al., 2017*), the number of sites associated with *IFNL4* genotype increased from 11 to 42 at a 5% FDR. One of the 11 previously reported associations was not reproduced (position 2576) as it was not tested in this study, due to more stringent frequency filtering (an amino acid had to be present in at least 20 samples to be tested). Thus, we have identified a further 32 associated sites at a 5% FDR compared to our previous report. There are two key factors that have contributed to this increased number of associated sites. Firstly, only HCV genotype 3a sequences found in those of white ancestry have been included in the analysis; our previous report included HCV genotype 2 and other genotype 3 subtypes from a broader ethnic mix. Secondly, we have used logistic regression and accounted for population structure by including viral and host genetic PCs as covariates in the analysis. This approach has recently been shown to be more powerful to detect associations in genome-to-genome analysis (*Naret et al., 2018*). Thus, focusing on a homogenous population (white ethnicity infected with genotype 3a) and using logistic regression has increased the power of our study to detect many more associations between the *IFNL4* locus and amino acid variation in the HCV polyprotein.

To distinguish the effect of *IFNL4* SNP rs12979860 on viral amino acid variation from nucleotide sequence variability, we investigated the association of the host *IFNL4* SNP rs12979860 with synonymous and non-synonymous codon changes in the HCV genome. *IFNL4* SNP rs12979860 genotypes were associated with 35 non-synonymous codon changes at a 10% FDR, but we also observed four significant associations with synonymous codon changes. This indicated that in addition to a widespread impact at the amino acid level, the *IFNL4* locus may also independently drive nucleotide diversity but to a lesser extent.

To further explore the impact of the *IFNL4* variants on HCV genomic sequences, we investigated viral dinucleotide frequencies. The HCV UpA dinucleotide frequency was significantly associated with the *IFNL4* SNP rs12979860 genotypes; interestingly, ribonuclease L (RNase-L), an ISG that cleaves viral RNA to control viral infections in animals (*Ding and Voinnet, 2007*), targets both UpA and UpU dinucleotides (*Wreschner et al., 1981*). Moreover, HCV genotype 1, which is relatively resistant to IFN-based therapy, has a lower frequency of UpA and UpU dinucleotides than the more IFN-sensitive HCV genotypes 2 and 3 (*Dao Thi et al., 2012*; *Kong et al., 2012*). Indeed, the UpA dinucleotide is targeted by RNA-degrading enzymes and its presence in a RNA sequence accelerates its degradation in the cytoplasm (*Simmonds et al., 2013*). It is possible that the more stringent immune environment of non-CC patients (with higher hepatic expression of ISGs) selects virus with a lower UpA frequency. However, we note that the *IFNL4* SNP rs12979860 non-CC patients have a modest reduction (0.9%) in their viral UpA frequencies relative to the CC patients and that this reduction could be mediated by the widespread amino acid changes associated with the *IFNL4* SNP rs12979860.

Previous studies have shown that IFN-λ4-P70 and IFN-λ4-S70 variants of IFN-λ4 protein have distinct phenotypes, both in vivo and in vitro. We hypothesized that if IFN-λ4 contributes to changes in the viral polyprotein, then the IFN-λ4-P70 and IFN-λ4-S70 variants should have different effect sizes on HCV-encoded amino acids and viral load. By imputing and phasing *IFNL4* SNPs in our cohort, we

inferred the haplotypes consisting of the *IFNL4* rs368234815 (ΔG/TT) and rs117648444 (G/A) variants. Using these data, we observed that the haplotype coding for the IFN-λ4-S70 variant had a smaller effect on viral amino acid variability compared to the haplotype coding for the IFN-λ4-P70 variant. This observation agrees with previous independent studies showing the reduced antiviral activity for IFN-λ4-S70 in vitro (*Bamford et al., 2018*; *Terczyńska-Dyla et al., 2014*). Moreover, the mean viral load in IFN-λ4-Null carrying patients was similar to those carrying the IFN-λ4-S70 variant; by contrast, those carrying an IFN-λ4-P70 variant had a reduced viral load, which correlates with its higher antiviral activity. Taken together, these observations reinforce the hypothesis that *IFNL4* is a functional gene with a major role in HCV infection. We conclude that production of IFN-λ4 protein drives an altered immune response that mediates reduced viral load with a broad impact on viral amino acid diversity.

In this study, we demonstrated by using paired host-HCV genomic data that host *IFNL4* gene, a cytokine that is part of the innate immune response and not expected to target specific viral residues, can mediate selection of amino acids at specific sites on the HCV polyprotein. We report that 1.4% (42/3021) of the HCV amino acids across the viral polyprotein are associated with host *IFNL4* SNP rs12979860 at a 5% FDR and that the impact on amino acid variation is spread across all viral proteins. In comparison, we previously reported that 0.7% of the HCV amino acids were associated with *HLA* class I and II alleles (*Ansari et al., 2017*) at a 5% FDR. The only other major driver of HCV amino acid variation is the host B cell response, which is largely restricted to modifying amino acids in the envelope glycoproteins, in particular E2 (*Ball et al., 2014*). Thus, although both arms of the adaptive immune system direct amino acid selection through pressure on epitopes recognised by T and B cell responses to infection, our results show that the innate immune response also has the capacity to drive particular polymorphisms in HCV-encoded proteins and that its impact is comparable to that of the T cell response (as indicated by HCV amino acid variants associated with HLA alleles).

Given that we did not observe any significant enhancement or depletion of association signals in a specific viral protein or in HLA-restricted epitopes, IFN-λ4 may exert its impact through a previously unknown mechanism or at more than one stage of the virus life cycle. We anticipate that this would result from distinct host responses in those who carry genetic variants that lead to IFN-λ4 synthesis as compared to individuals who carry the pseudogenized form of the gene. In common with other IFNs, IFN-λ4 induces a large number of ISGs, many of which are largely unstudied or poorly characterised. Within such an environment, it is possible that subtle selection of certain amino acids along the polyprotein will provide an advantage for viral entry, RNA replication, virion assembly and release.

Although there was no significant enrichment of associations comparing the structural with the non-structural proteins, the E2 glycoprotein contained the highest proportion of sites affected by *IFNL4* SNP rs12979860. From mapping these sites onto previously known functional domains on E2, we found that many residues were located either in hypervariable region 1 (HVR1) or on the surface of the protein. Indeed, some sites coincided with epitopes that are targets for the antibody response or have a role in virus entry (Appendix 1 and *Figure 1—figure supplement 5*). Since the host response to HCV genotype 3a infection induces pathways including those affecting B cell development (*Robinson et al., 2015*), we do not exclude the possibility that *IFNL4* SNP rs12979860 genotype either influences B cell response to infection or the process of virus binding and entry. Indeed, a recent report has demonstrated the emergence of variants in E2 during establishment of chronic infection that give enhanced resistance to interferon-induced transmembrane (IFITM) proteins (*Wrensch et al., 2019*). One of the E2 residues that encoded a different amino acid as chronicity developed was at position 500, which is an *IFNL4*-associated site (*Supplementary file 2*). Given that IFITM proteins potently inhibit virus entry, it is possible that differences in IFITM induction between those who do and do not produce IFN-λ4 may account for some of the footprint observed in the E2 protein sequence. Moreover, IFITM proteins cooperate with anti-HCV neutralising antibodies to enhance restriction of virus entry. Therefore, there may be interaction between genes differentially regulated between IFN-λ4 producers and non-producers and components of the adaptive immune system, which together influence the amino acids encoded at certain *IFNL4*-associated sites (*Wrensch et al., 2019*).

To reduce any confounding effects due to population stratification, we limited our analysis to a homogenous population of self-reported white origin infected with HCV genotype 3. Thus, our observations cannot directly be extended to other human populations or other HCV genotypes. This study provides a foundation for future analysis on whether *IFNL4* genetic variation also drives diversity in other HCV genotypes and subtypes across other ethnic populations (also see 'Adaptation of hepatitis C virus to interferon lambda polymorphism across multiple viral genotypes' by Chaturvedi *et al.* in this issue). Additionally, our observations were performed with in vivo data; further functional studies using appropriate in vitro model systems are needed to understand how the *IFNL4* locus drives HCV amino acid variation and modulates viral load. Such studies may also help to inform the basis for diversity and evolution of HCV in the presence or absence of *IFNL4*.

There are now multiple studies suggesting that the *IFNL3-IFNL4* locus could be a key player in the defense against viruses other than HCV. In HIV-infected patients, the rs368234815 variant has been associated with long-term non-progressor HIV-1 controllers (*Dominguez-Molina et al., 2017*). In influenza virus infection, *IFNL4* SNP rs8099917 was associated with increased sero-conversion after influenza vaccination (*Egli et al., 2014a*). *IFNL4* variants have also been associated with bronchiolitis (*Scagnolari et al., 2012*), cytomegalovirus (*Egli et al., 2014b*) and Andes virus (*Angulo et al., 2015*) infections. These observations suggest that *IFNL4* possibly plays a role in many viral infections and immune related diseases in the liver and other organs. Investigating how IFN-λ4 (a cytokine without epitope specificity) drives amino acid selectivity in the HCV polyprotein would add a new dimension to how the human innate immune system interacts with viruses and controls infectious diseases.

## Materials and methods

### Patient cohorts

For this study, we used patient data from the BOSON and EAP cohorts that have been described elsewhere (*Foster et al., 2015*; *Foster et al., 2016*). All patients provided written informed consent before undertaking any study-related procedures. The BOSON study protocol was approved by each institution's review board or ethics committee before study initiation. The study was conducted in accordance with the International Conference on Harmonisation Good Clinical Practice Guidelines and the Declaration of Helsinki (clinical trial registration number: NCT01962441). The EAP study conforms to the ethical guidelines of the 1975 Declaration of Helsinki as reflected in a priori approval by the institution's human research committee. The EAP patients were enrolled by consent into the HCV Research UK registry. Ethics approval for HCV Research UK was given by NRES Committee East Midlands - Derby 1 (Research Ethics Committee reference 11/EM/0314).

Patients from the BOSON cohort were recruited in five different countries (Australia, Canada, New Zealand, United Kingdom and United States). Patients from the EAP cohort were recruited exclusively in the United Kingdom.

To limit the potential impact of population structure, we restricted the analysis to patients of self-reported white ancestry infected with HCV genotype 3a for which we had obtained both host genome-wide SNP data and full-length HCV genome sequences. In total we included 485 patients in the study, 411 from the BOSON cohort and 74 from the EAP cohort.

The majority of the patients from the BOSON cohort have no or mild liver disease (compensated liver cirrhosis). The EAP cohort on the other hand consists of HCV-infected patients with advanced liver disease, the majority of whom had decompensated cirrhosis.

### Host genotyping and imputation

Informed consent for genetic analysis was obtained from all patients. Genomic DNA was extracted from buffy coat using Maxwell RSC Buffy Coat DNA Kit (Promega) as per the manufacturer's protocol and quantified using Qubit (Thermofisher). DNA samples from patients were genotyped using the Affymetrix UK Biobank array, as described elsewhere (*Ansari et al., 2017*). Phasing and imputation was performed using SHAPEIT2 (*Delaneau et al., 2013*)*Dao Thi et al., 2011* and IMPUTE2 (*Howie et al., 2009*) version 2.3.1 using default settings and the 1000 Genomes Phase III dataset as a reference population (*Auton et al., 2015*). Imputation quality was high for both rs117648444 and rs368234815 variants (information 0.974 and 0.994 respectively and certainty 0.995

and 0.997 respectively). Patients from the EAP cohort from whom enough DNA was available (62/74) were also independently genotyped for both rs117648444 and rs368234815 variants. Genotyping of *IFNL4* rs368234815 and rs117648444 was performed on DNA using the TaqMan SNP genotyping assay and sequences described previously (*Prokunina-Olsson et al., 2013*) with Type-it Fast SNP Probe PCR Master Mix (Qiagen). The concordance between genotyped and imputed genotypes was 100% for both variants.

## Virus sequencing

The generation and assembly of viral sequences from HCV-infected clinical samples for the BOSON and EAP cohorts have been described previously (*Ansari et al., 2017*; *Singer et al., 2019*).

## Statistical analysis

### Human and viral population structure

For the viral data, principal component analysis (PCA) was performed on the nucleotide data as follows. The presence and absence of each viral nucleotide at all variable sites in the alignment was coded as a binary variable such that a bi-allelic site on the viral genome was converted into two binary variables (one for each nucleotide), a tri-allelic site into three binary variables and a quad-allelic site into four binary variables. R (version 3.4.3, https://www.r-project.org) was then used to perform the PCA using the prcomp function with default settings. PCA was performed using flashpca (*Abraham and Inouye, 2014*) for human genotype data.

Whole-genome viral consensus sequences for each patient were aligned using MAFFT (*Katoh and Standley, 2013*) with default settings. This alignment was used to create a maximum-likelihood tree using RAxML (*Stamatakis, 2014*), assuming a general time-reversible model of nucleotide substitution under the gamma model of rate heterogeneity. The resulting tree was rooted at midpoint.

We used treeBreaker software (*Ansari and Didelot, 2016*) (https://github.com/ansariazim/treeBreaker) to measure the association between the virus phylogenetic tree and the host *IFNL4* SNP (CC vs. non-CC) and with the cohort from which the viral sequence was obtained (BOSON vs. EAP). This software uses a Bayesian model to infer whether the phenotype of interest is randomly distributed on the tips of the tree and to estimate which clades, if any, have a distinct distribution of the phenotype of interest from the rest of the tree. This software also performs Bayesian model comparison. It provides a Bayes factor for the alternative model (there is at least one or more branches with distinct distribution of the phenotype of interest) to the null model (there are no branches with distinct phenotype distribution on the tree). Bayes factor is a summary of the evidence provided by the data in favour of one model compared to another. In other words, the higher the Bayes factor, the more support there is for one model against another. Assuming that we are testing an alternative model against a null model, it has been suggested that the Bayes factor can be divided into four categories (*Kass and Raftery, 1995*). Bayes factor: 1 to 3.2, very little evidence against the null model. Bayes factor: 3.2 to 10, substantial evidence against null model and in favour of the alternative model. Bayes factor: 10 to 100, strong evidence against null model and in favour of the alternative model. Bayes factor: >100, decisive evidence against null model and in favour of the alternative model.

### Association tests

The univariate association between the *IFNL4* SNP rs12979860 (CC vs. non-CC) and the viral PCs was tested using logistic regression in R. We used the qvalue function from the qvalue package in R to perform the FDR analysis.

To choose 500 SNPs across the human genome with minor allele frequency (MAF) similar to the *IFNL4* SNP rs12979860 MAF, we used Fisher's exact test to compare all SNPs against the *IFNL4* SNP rs12979860 ($2 \times 3$ contingency table where the columns indicate the number of 0, 1 and 2 copies of the minor allele and the rows indicate the *IFNL4* SNP and the target SNP counts) and chose the 500 SNPs with the largest P-values (least significance). SNPs in the *IFNL3-IFNL4* region were not included.

To test for association between the virus amino acids and the host SNPs we used logistic regression in R including as covariates the first three host and the first two viral PCs. We investigated presence and absence of each amino acid at all variable sites, given that the amino acid was present in at least

20 HCV sequences in our dataset. The presence and absence of the viral amino acid was used as the response variable and the host SNP coded as 0 (homozygous for major alleles) and 1 (all other genotypes) as the explanatory variable (the same coding as the *IFNL4* SNP rs12979860 CC vs. non-CC).

To test for enrichment or depletion of the association signals in a viral protein or the epitope regions, we used Fisher's exact test. Each tested site is either within the target region or not and it is either classified as significant or not. The resulting 2 × 2 contingency table was tested using fisher.test function in R.

## Codon level analysis

To separate the impact of the *IFNL4* SNP rs12979860 on amino acids from nucleotides we investigated the nucleotide sequences at the codon level. At each codon site (where the most common codon had at least 20 synonymous and 20 non-synonymous codons) we used logistic regression to test for association between *IFNL4* SNP rs12979860 (CC vs. non-CC) and the changes from the most common codon to synonymous and non-synonymous codons. The *IFNL4* SNP rs12979860 was denoted as the response variable and the codons were used as a categorical explanatory variable with three levels. The effect sizes (log(OR)) and P-values were estimated for the synonymous and non-synonymous codons relative to the most common codon. We included the first two viral PCs and the first three host PCs as covariates in this analysis.

## Di-nucleotides analysis

To estimate the viral dinucleotide frequencies, the observed proportion of each dinucleotide was normalised by its expected proportion (assuming the nucleotides are independent the expected proportion can be calculated by multiplying the observed proportions for the relevant nucleotides). To test for association with *IFNL4* SNP rs12979860 genotype we used a linear regression where the normalised dinucleotide proportions were used as the response variable and the *IFNL4* SNP rs12979860 genotype as a categorical explanatory variable. We included the first two viral PCs and the first three host PCs as covariates.

## Effect of the three IFN-λ4 protein variants

To estimate the effect of the IFN-λ4 protein variants on the encoded HCV amino acids, we used the 76 sites associated with *IFNL4* SNP rs12979860 at a 10% FDR. HCV-infected patients were classified into three groups according to their predicted ability to produce IFN-λ4 protein: (i) no IFN-λ4 (two allelic copies of IFN-λ4-Null, $N_{BOSON}$ = 145, $N_{EAP}$ = 41), (ii) IFN-λ4–S70 (two copies of IFN-λ4-S70 or one copy of IFN-λ4-S70 and one copy of IFN-λ4-Null, $N_{BOSON}$ = 48, $N_{EAP}$ = 7), and (iii) IFN-λ4-P70 (at least one copy of IFN-λ4-P70, $N_{BOSON}$ = 218, $N_{EAP}$ = 26). We then used logistic regression to estimate the effect sizes (log(OR)) for IFN-λ4-P70 and IFN-λ4-S70 on the virus amino acids relative to IFN-λ4-Null. The presence and absence of the reported viral amino acid was used as the response variable and the host IFN-λ4 status was used as the explanatory variable with the IFN-λ4-Null used as the base level and the log(OR) for IFN-λ4-P70 and IFN-λ4-S70 were estimated relative to the IFN-λ4-Null base level. We included the first two viral PCs and the first three host PCs as covariates to account for host-virus populations co-structuring.

To test whether the effect sizes of IFN-λ4-P70 and IFN-λ4-S70 on viral amino acids are the same, we used the above estimated effect sizes and fitted a linear regression line to it. One viral site (position 2371) was excluded from this analysis as it had unreliable effect size estimate (log(OR) = −17) for IFN-λ4-S70 variant. Under the null hypothesis that IFN-λ4-P70 and IFN-λ4-S70 have the same effect sizes, we would expect that the linear regression line to have a slope of one. To test whether the slope of the fitted line is different from one, we used R to fit a linear regression line with intercept of zero. We used bootstrapping to account for the uncertainty associated with the estimated effect sizes of IFN-λ4-P70 and IFN-λ4-S70 on viral amino acids. We simulated 10,000 bootstrap datasets where the effect sizes of each IFN-λ4 variant on each HCV amino acid were simulated using a normal distribution with mean set to the estimated effect size of the variant on that HCV amino acid and standard deviation set to the standard error of the estimate. For each dataset we fitted a linear regression with intercept of zero and estimated the slope of the fit. The empirical bootstrap 95% confidence interval of the slope of the line was estimated as [2* best_estimate_slope – 97.5%_quantile_of_bootstrap_slopes, 2*

best_estimate_slope – 2.5%_quantile_of_bootstrap_slopes]. The 'best_estimate_slope' is the slope of the line estimated from the effect sizes without accounting for uncertainty.

To assess whether the mean viral load was different in the three patient groups of IFN-λ4-Null, IFN-λ4-P70 and IFN-λ4-S70, we used a Bayesian framework to perform model comparison (see Appendix 1 for further details). The models we considered comprised fixed and independent effects between the IFN-λ4 variants. We standardised the log10(viral load) so that it had a mean of zero and standard deviation of one. We used linear regression to get maximum likelihood estimates of the effects of IFN-λ4-S70 and IFN-λ4-P70 variants relative to the IFN-λ4-Null variant. The estimates were adjusted for cirrhosis status and population structure including the first two viral PCs and the first three host PCs in the regression as covariates. For each effect size, we assumed a normally distributed prior on the log(OR) of association with mean of zero. The prior covariance matrix determined the prior model assumptions. The elements of the covariance matrix were chosen such that the relevant prior model was set (see Appendix 1 for details).

To assess the evidence for interaction between host IFN-λ4 variants and viral amino acid site 2414, we used the same Bayesian framework detailed above. The patients were grouped into six categories based on the host IFN-λ4 variants and the presence or absence of serine at viral site 2414. We standardised the log10(viral load) so that it had a mean of zero and standard deviation of one. We used linear regression to get maximum likelihood estimates of the effects of 'IFN-λ4-Null + 2414 not serine', 'IFN-λ4-P70 + 2414 not serine', 'IFN-λ4-P70 + 2414 serine', 'IFN-λ4-S70 + 2414 not serine', 'IFN-λ4-S70 + 2414 serine' groups relative to the 'IFN-λ4-Null + 2414 serine' group. The estimates were adjusted for cirrhosis status and population structure including the first two viral PCs and the first three host PCs in the regression as covariates. The prior covariance matrix determined the prior model assumptions. The elements of the covariance matrix were chosen such that the relevant prior model was set (see Appendix 1 for details).

## Materials and correspondence

Correspondence and material requests should be addressed by contacting STOP-HCV http://www.stop-hcv.ox.ac.uk/contact.

## Acknowledgements

The authors thank Gilead Sciences for the provision of samples and data from the BOSON clinical study for use in these analyses and HCV Research UK (funded by the Medical Research Foundation [C0365]) for their assistance in handling and coordinating the release of samples for these analyses. The authors also thank Daniel J Wilson and Jacques Fellay for helpful comments. This work was funded by a grant from the Medical Research Council (MR/K01532X/1 – STOP-HCV Consortium). The work was supported by Core funding to the Wellcome Trust Centre for Human Genetics provided by the Wellcome Trust (090532/Z/09/Z). ECT is funded by Wellcome Trust as a clinical fellow (102789/Z/13/Z). EB is funded by the MRC as an MRC Senior Clinical Fellow with additional support from the Oxford NHIR BRC as a principal fellow. Professor Barnes is a National Institute for Health Research (NIHR) Senior Investigator. PK is funded by the Oxford Martin School, NIHR Biomedical Research Centre, Oxford, by the Wellcome Trust (109965MA) and NIH (U19AI082630). CCAS is funded by the Wellcome Trust (097364/Z/11/Z). Work in AHP and JM's laboratories are supported by MRC Core funding to the MRC-University of Glasgow Centre for Virus Research (MC_UU 12014/2). The views expressed in this article are those of the author(s) and not necessarily those of the NHS, the NIHR, or the Department of Health.

## Additional information

### Group author details

**STOP-HCV Consortium**
**J Ball**: University of Nottingham; **E Barnes**: University of Oxford; **G Burgess**: Conatus Pharmaceuticals; **G Cooke**: Imperial College London; **J Dillon**: University of Dundee; **G Foster**: Queen Mary University of London; **C Gore**: The Hepatitis C Trust; **N Guha**: University of

Nottingham; **R Halford**: The Hepatitis C Trust; **C Holmes**: University of Oxford; **E Hudson**: University of Oxford; **S Hutchinson**: Glasgow Caledonian University; **W Irving**: University of Nottingham; **S Khakoo**: University of Southampton; **P Klenerman**: University of Oxford; **N Martin**: University of Bristol; **T Mbisa**: Public Health England; **J McKeating**: University of Oxford; **J McLauchlan**: University of Glasgow; **A Miners**: London School of Hygiene and Tropical Medicine; **A Murray**: OncImmune; **P Shaw**: Merck; **P Simmonds**: University of Oxford; **S Smith**: The Hepatitis C Trust; **C Spencer**: University of Oxford; **E Thomson**: University of Glasgow; **P Troke**: Gilead Sciences; **P Vickerman**: University of Bristol; **N Zitzmann**: University of Oxford

## Competing interests
Graham R Foster: Grants Consulting and Speaker/Advisory Board: AbbVie, Alcura, Bristol-Myers Squibb, Gilead, Janssen, GlaxoSmithKline, Merck, Roche, Springbank, Idenix, Tekmira, Novartis. William L Irving: Grants, Consulting and Advisory/ Speaker Board: Roche, Janssen Cilag, Gilead Sciences, Novartis, GlaxoSmithKline, Pfizer, Abbvie and Bristol-Myers Squibb. Kosh Agarwal: Grants, Consulting and Advisory/ Speaker Board: Achillion, Alnylam, Astellas, Abbvie, Bristol-Myers Squibb, Gilead, GlaxoSmithKline, Janssen, Merck, Roche, Novartis, Vir. The other authors declare that no competing interests exist.

## Funding

| Funder | Grant reference number | Author |
|---|---|---|
| Medical Research Council | MR/K01532X/1 - STOP-HCV Consortium | M Azim Ansari<br>Elihu Aranday-Cortes<br>Camilla LC Ip<br>Ana da Silva Filipe<br>Siu Hin Lau<br>Connor Bamford<br>David Bonsall<br>Amy Trebes<br>Paolo Piazza<br>Vattipally Sreenu<br>Vanessa M Cowton<br>Emma Hudson<br>Rory Bowden<br>Arvind H Patel<br>Graham R Foster<br>William L Irving<br>Kosh Agarwal<br>Emma C Thomson<br>Peter Simmonds<br>Paul Klenerman<br>Chris Holmes<br>Eleanor Barnes<br>Chris CA Spencer<br>John McLauchlan<br>Vincent Pedergnana |
| National Institute for Health Research | U19AI082630 | Paul Klenerman |
| Oxford Martin School, University of Oxford | 109965MA | Paul Klenerman |
| Wellcome | 090532/Z/09/Z | M Azim Ansari<br>Camilla LC Ip<br>David Bonsall<br>Amy Trebes<br>Paolo Piazza<br>Rory Bowden<br>Chris CA Spencer<br>Vincent Pedergnana |
| Wellcome | 097364/Z/11/Z | Chris CA Spencer |
| Wellcome | 102789/Z/13/Z | Emma C Thomson |

| Medical Research Council | MC_UU 12014/2 | Elihu Aranday-Cortes |
| | | Ana da Silva Filipe |
| | | Siu Hin Lau |
| | | Connor Bamford |
| | | Vattipally Sreenu |
| | | Vanessa M Cowton |
| | | Arvind H Patel |
| | | John McLauchlan |

The funders had no role in study design, data collection and interpretation, or the decision to submit the work for publication.

## Author contributions

M Azim Ansari, Conceptualization; Data curation; Software; Formal analysis; Investigation; Methodology; Writing—original draft; Writing—review and editing; Elihu Aranday-Cortes, Conceptualization; Data curation; Formal analysis; Validation; Writing—original draft; Writing—review and editing; Camilla LC Ip, Data curation; Software; Methodology; Writing—review and editing; Ana da Silva Filipe, Connor Bamford, Resources; Data curation; Formal analysis; Writing—review and editing; Siu Hin Lau, Data curation; Formal analysis; Writing—review and editing; David Bonsall, Amy Trebes, Paolo Piazza, Vattipally Sreenu, Resources; Software; Writing—review and editing; Vanessa M Cowton, Resources; Formal analysis; Investigation; Writing—review and editing; STOP-HCV Consortium, Resources; Investigation; Funding acquisition; Writing- review and editing; Emma Hudson, Funding acquisition; Project administration; Writing—review and editing; Rory Bowden, Conceptualization; Resources; Software; Methodology; Writing—review and editing; Arvind H Patel, Resources; Investigation; Writing—review and editing; Graham R Foster, Resources; Validation; Writing—review and editing; William L Irving, Paul Klenerman, Resources; Funding acquisition; Writing—review and editing; Kosh Agarwal, Emma C Thomson, Resources; Writing—review and editing; Peter Simmonds, Supervision; Funding acquisition; Writing—review and editing; Chris Holmes, Methodology; Writing—review and editing; Eleanor Barnes, Funding acquisition; Writing—original draft; Project administration; Writing—review; and editing; Chris CA Spencer, Conceptualization; Data curation; Formal analysis; Funding acquisition; Investigation; Methodology; Writing—original draft; Writing—review and editing; John McLauchlan, Conceptualization; Supervision; Funding acquisition; Validation; Writing—original draft; Writing—review and editing; Vincent Pedergnana, Conceptualization; Data curation; Formal analysis; Supervision; Validation; Methodology; Writing—original draft; Writing—review and editing

## Author ORCIDs

Elihu Aranday-Cortes https://orcid.org/0000-0003-2637-2307
Siu Hin Lau https://orcid.org/0000-0003-4866-8248
Connor Bamford https://orcid.org/0000-0003-4895-1983
Rory Bowden https://orcid.org/0000-0001-8596-0366
William L Irving https://orcid.org/0000-0002-7268-3168
Paul Klenerman https://orcid.org/0000-0003-4307-9161
Vincent Pedergnana https://orcid.org/0000-0002-7852-5339

## Ethics

Human subjects: The Boson study was conducted at 80 sites in the United Kingdom, Australia, the United States, Canada and New Zealand. All patients provided written informed consent before undertaking any study-related procedures. The study protocol was approved by each institution's review board or ethics committee before study initiation. The study was conducted in accordance with the International Conference on Harmonization Good Clinical Practice Guidelines and the Declaration of Helsinki (clinical trial registration number: NCT01962441). The EAP study conforms to the ethical guidelines of the 1975 Declaration of Helsinki as reflected in a priori approval by the institution's human research committee. The EAP patients were enrolled by consent into the HCV Research UK registry. Ethics approval for HCV Research UK was given by NRES Committee East Midlands - Derby 1 (Research Ethics Committee reference 11/EM/0314). Informed consent for genetic analysis was obtained from all patients.

Decision letter and Author response
Decision letter https://doi.org/10.7554/eLife.42463.026
Author response https://doi.org/10.7554/eLife.42463.027

## Additional files

### Supplementary files

• Supplementary file 1. Demographic, clinical and genetic characteristics of the BOSON and EAP cohorts.
DOI: https://doi.org/10.7554/eLife.42463.014

• Supplementary file 2. Host *IFNL4* SNP rs12979860 association with HCV amino acids at a 10% FDR. We used logistic regression to test for association between host *IFNL4* SNP (CC vs. non-CC) and presence and absence of amino acids. We included the first two viral and the first three host PCs as covariates. Only amino acids that were present in at least 20 samples were tested (977 amino acids in 471 sites). For each associated site, we have reported all amino acid with a count of >= 20 in reducing frequency order and highlighted the most associated amino acid in bold. The amino acid frequency in CC and nonCC patients, the P-value, log(OR), the standard error and q-value are reported for the most associated amino acid at the site.
DOI: https://doi.org/10.7554/eLife.42463.015

• Supplementary file 3. P-value of Fisher's exact test for enrichment or depletion of the association signals in HCV proteins and HLA-restricted epitopes.
DOI: https://doi.org/10.7554/eLife.42463.016

• Supplementary file 4. Associations between HCV amino acids and the 500 frequency-matched host SNPs at a 5% FDR. Note that the FDR was calculated independently for each viral GWAS against each host SNP. All the significant associations for each viral GWAS (against each of the 500 frequency matched host SNP) are shown in this table.
DOI: https://doi.org/10.7554/eLife.42463.017

• Supplementary file 5. Host *IFNL4* SNP rs12979860 association with changes from the most common codon to non-synonymous codons in HCV at a 10% FDR. We used logistic regression to test for association between host *IFNL4* SNP (CC vs. non-CC) and codon changes. We included the first two viral and the first three host PCs as covariates. Only codons at which there were at least 20 synonymous and 20 non-synonymous codons for the most common codon at the site (348 codon sites across the HCV coding sequence) were included in the analysis.
DOI: https://doi.org/10.7554/eLife.42463.018

• Supplementary file 6. Host *IFNL4* SNP rs12979860 association with changes from the most common codon to synonymous codons in HCV at a 10% FDR. We used logistic regression to test for association between host *IFNL4* SNP (CC vs. non-CC) and codon changes. We included the first two viral and the first three host PCs as covariate. Only codons at which there were at least 20 synonymous and 20 non-synonymous codons for the most common codon at the site (348 codon sites across the HCV coding sequence) were included in the analysis.
DOI: https://doi.org/10.7554/eLife.42463.019

• Supplementary file 7. *IFNL4* haplotype combination and predicted protein for host SNPs rs117648444 and rs368234815 in the EAP (N = 74) and BOSON (N = 411) cohorts.
DOI: https://doi.org/10.7554/eLife.42463.020

• Transparent reporting form
DOI: https://doi.org/10.7554/eLife.42463.021

### Data availability

Human genotype data underlying this manuscript are deposited in the European Genome-phenome Archive under accession code EGAS00001002324. HCV sequence data underlying this manuscript are deposited in GenBank under accession codes KY620313-KY620880. Information on access to the study data is available at http://www.stop-hcv.ox.ac.uk/data-access.

The following previously published dataset was used:

| Author(s) | Year | Dataset title | Dataset URL | Database and Identifier |
|---|---|---|---|---|
| Ansari MA, Pedergnana V, Ip CLC, Magri A, Von Delft A, Bonsall D, Chaturvedi N, Bartha I, Smith D, Nicholson G, McVean G, Trebes A, Piazza P, Fellay J, Cooke G, Foster GR, STOP-HCV Consortium, Hudson E, McLauchlan J, Simmonds P, Bowden R, Klenerman P, Barnes E, Spencer CCA | 2017 | BOSON | https://www.ebi.ac.uk/ega/studies/EGAS00001002324 | European Genome-phenome Archive, EGAS00001002324 |

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

## Appendix 1

DOI: https://doi.org/10.7554/eLife.42463.022

## Viral principal components association with host *IFNL4* SNP rs12979860

The first two viral PCs were clustered with clades on the virus phylogeny (*Figure 1—figure supplement 2a*). The other PCs changed more gradually and were grouped to a lesser degree with specific clades on the tree. To explore any specific role for *IFNL4* SNP rs12979860 on viral diversity, we tested whether any viral PCs were associated with SNP rs12979860 genotypes in a univariate analysis and observed that the fifth and seventh PCs were associated with *IFNL4* SNP rs12979860 genotypes (*Figure 1—figure supplement 2b–d*) respectively explaining 0.7% and 0.5% of the total variance in the nucleotide sequences (*Figure 1—figure supplement 1b*).

To investigate whether the observed associations between *IFNL4* SNP rs12979860 genotypes and viral PCs were due to host-virus population co-structuring, we selected 500 SNPs across the human genome with frequencies similar to the *IFNL4* SNP rs12979860 SNP and tested for association between these SNPs and the viral PCs (*Figure 1—figure supplement 2b*). If host-virus populations co-structuring was responsible for the observed association between viral PCs and host *IFNL4* SNP rs12979860 genotypes, we would expect these 500 SNPs to also correlate with fifth and seventh viral PCs. At a 10% FDR, none of the viral PCs were associated with any of the 500 frequency-matched SNPs. Overall, these results indicate that the observed association between *IFNL4* SNP rs12979860 genotypes and the two viral PCs is a consequence of biological interaction between the *IFNL4* locus and the virus sequences and not due to population structure or other systematic bias.

## Amino acids associated with *IFNL4* SNP rs12979860 genotypes within the viral protein E2

Twenty-six of the 76 sites (34%) associated with the *IFNL4* locus at 10% FDR lie within the E2 glycoprotein that is involved in virus entry and directly interacts with two critical cell surface receptors, CD81 and SR-BI. Almost half of these IFN-λ4-associated residues are present in the solved E2 core structure. Interestingly, each of these amino acids is located on the surface of E2 (*Figure 1—figure supplement 5*) (*Kong et al., 2013*). Residues 438, 442, 521, 524 and 546 lie on the neutralizing face of E2; four are in regions targeted by neutralizing antibodies known as Epitope II (aa436-446) and Epitope III (aa523-549), which contain CD81-receptor binding residues. Five residues 576, 576d, 577, 578 and 580 are located within the intergenotypic variable region (igVR) that has a role in virion assembly (*Albecka et al., 2011*; *McCaffrey et al., 2011*). Recent evidence suggests that the igVR is under strong selective pressure during infection and that it may function to modulate exposure of both the CD81 binding region and antibody epitopes on the surface (*Alhammad et al., 2015*). This is supported by a study where mutation of igVR residues to alanine reduced binding of non-neutralizing antibodies (*Pierce et al., 2016*). For sites that are not present in the structure, eight lie within hypervariable region 1 (HVR1) (aa384-411), an immunodominant region of E2, and two are in hypervariable region 2 (HVR2) (aa474-482). Residue 399 in the HVR1 has been implicated in receptor interaction with SR-BI and heparan sulphate (*Dao Thi et al., 2011*; *Bartosch et al., 2005*). Remarkably, residues 457, 500, 501, 558 and 662 have all been shown to modulate the presentation of antibody epitopes as mutation to alanine altered antibody binding (*Pierce et al., 2016*). Residue 741 lies within the transmembrane domain (718-742).

While it is well-documented that type I IFNs can impact the B-cell response via direct and indirect mechanisms, there is little information regarding any influence by type III IFNs. Crucially, it has been shown that both naïve and memory B cells express the IFN-λ1 receptor (*de Groen et al., 2015*). Furthermore, there is evidence that type III interferon can influence antibody production (*de Groen et al., 2015*; *Egli et al., 2014a*). Together, our

analysis suggests that *IFNL4* may exert selection pressure on E2 modulating the structure such that presentation of receptor binding sites and antibody epitopes are altered.

## Viral dinucleotide frequencies in BOSON and EAP cohorts and their association with *IFNL4* SNP rs12979860.

To investigate how *IFNL4* impacts on viral sequences, we estimated the dinucleotide frequencies for each HCV sequence in the different *IFNL4* SNP rs12979860 groups (CC vs. non-CC, **Figure 1—figure supplements 7** and **8**) in the two cohorts. In the BOSON cohort, the UpA dinucleotide frequency (estimated as the ratio of the observed to expected frequency) was significantly lower in the *IFNL4* non-CC group compared to the CC group ($p=2.7\times10^{-6}$) using linear regression adjusting for two viral PCs and three host PCs. On the contrary, the UpG dinucleotide frequency was significantly higher in the *IFNL4* non-CC group compared to the CC group ($p=2.3\times10^{-5}$). The CpC and CpA dinucleotide frequencies were also significantly different between the *IFNL4* genotypes (CC vs. non-CC) using a Bonferroni correction for multiple testing ($p<0.003$). In the EAP cohort, the same trend, although not significant, was observed for the UpA and UpG dinucleotide frequencies.

## Bayesian model comparison for genetic effect sizes on viral load

We used approximate Bayes factors (ABFs) to estimate the posterior probability of each model of genetic association as described in **Band et al. (2013)**. The ABF differs from the Bayes Factor in that it depends on a normal approximation of the regression likelihood (up to a constant) as suggested by **Wakefield (2007)**. For each model of association we assume a prior on the effect size which is normally distributed around zero. By changing the prior covariance (or correlation) of effect sizes we can formally compare models. The ABF for each model is calculated as the ratio of the approximate marginal likelihood of that model to that of the null model where all the prior weight is on an effect size of zero; note that the ABF for the null model is then equal to 1. Under the assumption that exactly one of the models is correct and all models are equally likely a priori, the posterior probability of a given model is calculated as the ABF for that model divided by the sum of the ABFs for all models under consideration.

Calculation of the marginal likelihood requires specification of a prior distribution for the effects as well as maximum likelihood (ML) point estimates of these effects with their asymptotic standard errors. As a concrete example, let us use the IFN-λ4 variants effects on viral load. Using a linear regression we estimate the two effect sizes for IFN-λ4-P70 and IFN-λ4-S70 relative to the IFN-λ4-Null variant and their asymptomatic standard error. These estimates are adjusted for cirrhosis status and population structure by including the fist two viral PCs and the first three host PCs. We then define the covariance prior matrix which in this case is a $2 \times 2$ matrix indicating the correlation and variances of effect sizes of IFN-λ4-S70 and IFN-λ4-P70 relative to the IFN-λ4-Null variant. The five tested models and their priors are as follows:

1) Both IFN-λ4-S70 and IFN-λ4-P70 have effect sizes of zero (their effects on viral load are identical to the IFN-λ4-Null variant).

$$\Sigma_1 = \begin{bmatrix} 0 & 0 \\ 0 & 0 \end{bmatrix}$$

2) IFN-λ4-S70 has an independent effect and IFN-λ4-P70 has an effect size of zero (the effect on viral load is the same as the IFN-λ4-Null variant).

$$\Sigma_2 = \begin{bmatrix} 1 & 0 \\ 0 & 0 \end{bmatrix}$$

3) IFN-λ4-S70 has an effect size of zero (the effect on viral load is the same as the IFN-λ4-Null variant) and IFN-λ4-P70 has an independent effect size.

$$\Sigma_3 = \begin{bmatrix} 0 & 0 \\ 0 & 1 \end{bmatrix}$$

4) Both IFN-λ4-S70 and IFN-λ4-P70 have independent effect sizes (their effect on viral load are different from the IFN-λ4-Null variant and different from each other).

$$\Sigma_4 = \begin{bmatrix} 1 & 0 \\ 0 & 1 \end{bmatrix}$$

5) Both IFN-λ4-S70 and IFN-λ4-P70 have independent and identical effect sizes (their effects on viral load are the same but different from the IFN-λ4-Null variant).

$$\Sigma_5 = \begin{bmatrix} 1 & 1 \\ 1 & 1 \end{bmatrix}$$

The approximate marginal likelihood for any given choice of the prior model is then the value of the multivariate normal density at the ML estimate:

$$MVN\left( \begin{bmatrix} \hat{\beta}_1 \\ \hat{\beta}_2 \end{bmatrix}; \begin{bmatrix} 0 \\ 0 \end{bmatrix}, \begin{bmatrix} ccSE^2_{\beta_1} & 0 \\ 0 & SE^2_{\beta_2} \end{bmatrix} + \Sigma_i \right)$$

To investigate the interaction between *IFNL4* genotypes (CC vs. non-CC), viral load and site 2414 in the NS5A viral protein, we used the Bayesian approach detailed above. We divided the patients into six groups (the three variants of IFN-λ4 and presence or absence of serine at viral site 2414). We then fitted a linear regression with standardised log10(viral load) as the response variable and a categorical variable with the six levels as the explanatory variable. Five effect sizes and their standard error were estimated for the groups 'IFN-λ4-Null + 2414 not serine', 'IFN-λ4-P70 + 2414 not serine', 'IFN-λ4-P70 + 2414 serine', 'IFN-λ4-S70 + 2414 not serine', 'IFN-λ4-S70 + 2414 serine' relative to the 'IFN-λ4-Null + 2414 serine' group. We then used the prior covariance matrix (a 5 × 5 matrix) to define 58 models of association (from all effect sizes being zero to all being different). The posterior probability for each model was estimated as described above. Model five, where only the 'IFN-λ4-P70 + 2414 serine' group had an effect size different from the other groups was the most supported (posterior probability = 0.33). Twelve other models had posterior probabilities greater than their prior probabilities. In all these models, the 'IFN-λ4-P70 + 2414 serine' group had an effect size different from the base group. The marginal posterior probability of all the models in which the effect size of group 'IFN-λ4-P70 + 2414 serine' is different from the 'IFN-λ4-P70 + 2414 not serine' is 0.98. Furthermore the marginal posterior probability of all the models in which the effect size of the 'IFN-λ4-P70 + 2414 serine' is different from the 'IFN-λ4-S70 + 2414 serine' is 0.97.

