## [Decision Letter]

Thank you for submitting your article "Broad Impact of Interferon Lambda 4 on Hepatitis C Virus Diversity" for consideration by *eLife*. Your article has been reviewed by four peer reviewers, and the evaluation has been overseen by a Reviewing Editor and Wenhui Li as the Senior Editor. The reviewers have opted to remain anonymous.

The reviewers have discussed the reviews with one another and the Reviewing Editor has drafted this decision to help you prepare a revised submission.

Summary:

Previously, Ansari et al. reported results of a genome-to-genome study of 542 individuals who were chronically infected with HCV (predominantly viral genotype (VGT) 3, but also VGT 2; Ansari et al., 2017). Genotype for the *IFNL4* rs12979860 SNP marker associated with 11 amino acid polymorphisms on the HCV polyprotein (60, 109 [C]; 500, 501, 576d, 578, 741 [E2]; 2414 [NS5A]; 2570, 2576, 2991 [NS5B]) based on a 5% false discovery rate (FDR). The strongest association was for 2570 in NS5B; residue 2414 in NS5A associated with HCV RNA levels amongst patients infected with VGT 3a. The present paper, which is restricted to the 485 subjects who were infected with VGT 3a, contains important new data, but the analytical approach used to arrive at these conclusions is complicated and often confusing. A more focused and unified presentation of the results is needed.

Essential revisions:

Ansari et al. have now imputed genotype for *IFNL4* rs368234815, which controls generation of the IFN-λ4 protein, and *IFNL4* rs117648444, a non-synonymous polymorphism that defines a structural variant of the IFN-λ4 protein. Haplotypes comprised of these variants generate different versions of the IFN-λ4 protein. The authors report that variation in the HCV genome and HCV RNA levels associate with these different haplotypes, consistent with previous associations between these functional *IFNL4* haplotypes and HCV clearance. This finding is novel and potentially important, as it would provide additional support that the IFN activity of *IFNL4* affects the observed phenotype.

This finding raises some additional questions:

Re: the effect of the *IFNL4* loci on viral load, does that translate directly to a lower level of viral replication in patients with *IFNL4* P70? Might this be investigated from the diversity of the viral quasi-species found in the patient?

Re: the coupling between *IFNL4* P70 and S2414 in the HCV genome, is *IFNL4* P70 driving selection of a specific amino acid at this position?

In the Nature Genetics paper, *IFNL4* rs12979860 associated with 11 amino acid polymorphisms, but now, in a smaller cohort, that SNP apparently associates with 42 sites (both based on a 5% FDR). The authors should explain the reason for that striking difference and comment more generally about how findings from the current paper differ from the previous publication. Similarly, the paper should be clearer on which data and results are original to this paper with previously published data referenced to the Nature Genetics paper.

The value of including the Expanded Access Programme (EAP) subjects is unclear. EAP contributes only ~15% of the total subjects and differs from the BOSON group regarding important demographic and clinical characteristics (sex, prevalence of cirrhosis, HCV RNA levels), as well as genotype frequencies of rs12979860. If EAP subjects are retained in the revised paper, these differences should be considered in the analysis and addressed in the Discussion.

The reviewers have many questions and concerns about the statistical methods and the analyses.

Multiple testing: FDRs of either 5% or 20%, as well as Bonferroni correction are employed. Unless there are compelling reasons for these different approaches (which should be stated), a uniform approach to multiple testing adjustment should be used. The Abstract states rs12979860 genotype associated with 4% of viral amino acid sites across the HCV polyprotein. Based on the discussion (that finding does not appear in the results), this result is based on a 20% FDR, whereas findings presented in the Results (and the previous paper) are based on a 5% FDR. A 20% FDR seems very high and needs to be justified.

Genomic inflation factor for *IFNL4* rs12979860 and 500 SNPs with similar frequencies: The genome inflation factor (represented by λ) is used to examine assumptions re: cryptic relatedness when a large set of SNP markers are tested for association with a dichotomous trait in a GWAS (https://www.ncbi.nlm.nih.gov/pubmed/11315092 https://journals.plos.org/plosone/article?id=10.1371/journal.pone.0019416), but it is unclear what is being analyzed here – association of viral variants with a trait (yes or no host SNP)? Are the assumptions about the distribution of viral variants the same as for distribution of host germline variants, for which this approach was developed? This is not a standard approach to test for unaccounted population structure or other biases and the logic for doing this is unclear. Can the authors provide a reference to support use of this method?

A λ value of 2.16 is extremely high and indicates cryptic relatedness, but in this case that statistic is impossible to interpret it because the approach is not described adequately.

Principal component analysis: What is used for PCA in the host? There is no explanation and the plot differs from those used for GWAS, where study samples are plotted in relation to reference populations.

Which viral PCs were included? Were the PCs used as continuous variables in any models? (Note: "principal" is spelled as "principle" in several places.)

Assessment of Confounding, Interaction and Mediation: Assessment as to whether adjustment for potential confounders (e.g. sex, age, study [EAP or BOSON], cirrhosis) is needed. Were genotypes associated with any patient characteristics? E.g. age or cirrhosis status?

Stratified analyses should be performed to identify possible interactions for those variables, especially sex (interaction between *IFNL4* genotype and sex has been reported for associations with hepatic fibrosis).

The association of *IFNL4* genotype with the frequency of HCV polymorphisms could reflect an effect of *IFNL4* on viral replication rates. To assess that possibility, the investigators should compare the results of two logistic regression models: one that does and one that does not include HCV RNA as an additional covariate to *IFNL4*. Otherwise these paired models should include identical adjustments.

Other comments re: statistical analyses

Subsection “*IFNL4* SNP has a widespread impact on the viral amino acids” first paragraph: how was the "expected median" computed?

Subsection “*IFNL4* SNP has a widespread impact on the viral amino acids” second paragraph: was does "frequency matched" mean? With the same MAF?

Subsection “Subsection “*IFNL4* SNP has a widespread impact on the viral amino acids” first paragraph”, third paragraph: please state outcome variable for the logistic regression models.

Subsection “*IFNL4* SNP has a widespread impact on the viral amino acids”, third paragraph: please define FDR and give reference.

Subsection “Statistical analysis”, third paragraph: please state clearly that the SNP was the outcome for the logistic models.

Subsection “Statistical analysis”, eleventh paragraph: To obtain maximum likelihood estimates one needs to assume a normal distribution for log_10_(viral load) transformed data. IN my experience this assumption is not true for log_10_(viral load) transformed data. However, least squares estimates do not require this assumption.

Subsection “Statistical analysis”, eleventh paragraph: when a line was fit through the log(OR) estimates, how were the standard deviations of the log(ORs) used? That uncertainty needs to be accommodated.

Other general comments:

The presentation is hard to follow with most data presented as minimally annotated supplementary materials with limited legends provided separately. Providing more detailed legends next to corresponding figures and tables should make it easier to follow.

In the Discussion, the limitations of the study should be explored.

[Editors' note: further revisions were requested prior to acceptance, as described below.]

Thank you for resubmitting your work entitled "Interferon lambda 4 impacts on the genetic diversity of hepatitis C virus" for further consideration at *eLife*. Your revised article has been favorably evaluated by Wendy Garrett as the Senior Editor and a Reviewing Editor.

The manuscript has been improved but there are some remaining issues that need to be addressed before acceptance, as outlined below:

Overall, the authors were responsive to reviewer comments and the paper is much improved. The analytical approach remains complicated and the paper is still challenging to read. The authors should consider and address the following comments.

Multiple testing: The authors eliminated use of the Bonferroni correction and a false discovery rate (FDR) of 20%. The paper still presents two different FDR thresholds (5% and 10%) for many analyses and the reason for doing so is unclear. It would be simpler to report a single set of results based on a 5% FDR, the threshold used in the previous publication from this group.

Normality of viral load data: It is not clear from a visual inspection of the Q-Q plot that these data are normally distributed. A P-value for fit would be a more objective measure.

Second paragraph of the Introduction “substitutes a proline for a serine […]”: Terczyńska-Dyla et. al state, “an amino-acid substitution in the IFNλ4 protein changing a proline at position 70 to a serine (P70S) […]”. To this reader, that means a serine is substituted for a proline. Alternatively, the authors might use the language of Terczyńska-Dyla et. al to describe this variant.

Subsection “Host and virus genetic structures”: Without any explanation, it is unclear how to interpret the Bayes factors of 249 and 1.1.

How the patients are divided into *IFNL4*-null, S70 and P70 groupings could be clearer. Supplementary file 7 would present that information if the groups were arranged together and labeled.

Subsection “Host and virus genetic structures”, ninth paragraph – The co-submission by Chaturvedi et al. has been accepted for publication and might be referenced here.

---

## [Author Response]

Essential revisions:Ansari et al. have now imputed genotype for IFNL4 rs368234815, which controls generation of the IFN-λ4 protein, and IFNL4 rs117648444, a non-synonymous polymorphism that defines a structural variant of the IFN-λ4 protein. Haplotypes comprised of these variants generate different versions of the IFN-λ4 protein. The authors report that variation in the HCV genome and HCV RNA levels associate with these different haplotypes, consistent with previous associations between these functional IFNL4 haplotypes and HCV clearance. This finding is novel and potentially important, as it would provide additional support that the IFN activity of IFNL4 affects the observed phenotype.This finding raises some additional questions:Re: the effect of the IFNL4 loci on viral load, does that translate directly to a lower level of viral replication in patients with IFNL4 P70? Might this be investigated from the diversity of the viral quasi-species found in the patient?

We note the comment from the reviewers on whether the viral load translates directly to a lower level of viral replication. While lower viral replication (i.e. replication of viral RNA) could contribute to reduced viral load, we would not exclude a contribution from other parts of the viral life cycle, especially since the *IFNL4* footprint extends across both structural and non-structural proteins. Studies on the effects on viral RNA replication would require further analysis. However, during the review process, we are aware of a new report from Thomas Baumert’s group which reveals interplay between the innate immune response (IFITM proteins) and neutralization antibodies (Wrensch et al., Hepatology in press) that affect virus entry. Moreover, their study shows there is a change in the amino acids encoded by E2 from acute to chronic infection that generates IFITM-resistance. One of the sites that changes is at position 500 in the E2 protein, which is a position associated with *IFNL4* genotype in our study (see Figure 1—figure supplement 2). Thus there could be differences in the interferon response dependent on *IFNL4* genotype (i.e. producers vs. non-producers) that affects stages of the virus life cycle other than RNA replication (e.g. virus entry as in the Wrensch et al. paper or also translation, assembly and release). All of the above is addressed in the Discussion. As to the second point raised by the reviewer about whether lower viral load may influence the diversity of quasispecies, our analysis has not investigated the viral quasispecies in individuals (i.e. intra-host viral diversity); all of our analysis has been conducted on the consensus sequences and in this way, we have examined inter-host viral diversity. We agree with the reviewer that any reduction in viral load mediated by *IFNL4*-P70 could influence the quasispecies at the intra-host level. However, this would require a deeper analysis of the sequence data, preferably combined with functional in vitro studies.

Re: the coupling between IFNL4 P70 and S2414 in the HCV genome, is IFNL4 P70 driving selection of a specific amino acid at this position?

Since there is a significant association between serine at position 2414 and *IFNL4*-P70 variant, our interpretation would be that *IFNL4*-P70 does exert selection of S2414. However, this is potentially true also for the other associated sites that have statistical significance. This point is made in the revised Abstract (see final sentence). However, specifically in relation to S2414, we have added the following sentence to the paper: “The viral serine (S) residue at site 2414 (S2414) was significantly enriched in *IFNL4*-P70 carrying patients compared to *IFNL4*-S70 (P = 3.39x10^-03^) and *IFNL4*-Null (P = 5.94x10^-09^) carrying patients. S2414 was present in 86% (211/244) of IFNL-P70 carrying patients, 69% (38/55) of *IFNL4*-S70 carrying patients and 62% (114/185) of *IFNL4*-Null carrying patients.”

In the Nature Genetics paper, IFNL4 rs12979860 associated with 11 amino acid polymorphisms, but now, in a smaller cohort, that SNP apparently associates with 42 sites (both based on a 5% FDR). The authors should explain the reason for that striking difference and comment more generally about how findings from the current paper differ from the previous publication. Similarly, the paper should be clearer on which data and results are original to this paper with previously published data referenced to the Nature Genetics paper.

Our previous analysis in the Nature Genetics paper used 542 samples from individuals with different ethnic backgrounds (the two main ethnic groups were white [n=452] and Asian [n=74]). Additionally these individuals were infected by HCV genotypes 2 (n=45) and 3 (n=496) with different subtypes (gt3a, gt3b and gt3c). That study was aimed mainly at identifying any human genes that may be associated with viral polymorphisms. Not surprisingly, we identified viral variants that were linked to HLA alleles. Surprisingly, we found that *IFNL4* genotype was also associated with variants in the virus polyprotein. However, the depth of the study was limited by the mixture of ethnicities and viral genotypes/subtypes in the BOSON cohort. We believe that our current report does not simply confirm the findings in the Nature Genetics paper but expands and enhances the extent of the footprint of *IFNL4* genotype on variants in the viral polyprotein.

With regard to the increased number of associated sites, there are two major factors compared to our previous report that contribute to this enhancement. Firstly, in the current paper, we have limited the dataset to individuals with self-reported white ancestry infected with HCV gt3a to avoid potential bias due to co-structuring between host and virus populations. To increase the power of the study, we included 74 patients from another cohort, the EAP cohort, who were also of white ethnicity and infected with gt3a. This gave a total of 485 patients with identical ethnic background and infection with the same HCV genotype/subtype. As with any GWAS study, confidence in statistically significant associations is greater with more data and this was a major factor for inclusion of the EAP group. The EAP and BOSON cohorts do differ with regard to severity of liver disease; the EAP cohort consists largely of patients with decompensated cirrhosis while the BOSON cohort were recruited on the basis of milder disease. We did not find that differences in disease severity affected our analysis (see below). Therefore, the associations between *IFNL4* genotype and viral variants in gt3a apparently apply irrespective of the extent of liver disease.

The second important difference between the two studies is that in the Nature Genetics paper we used a different methodology to analyse the impact of SNP rs12979860 on the viral variants. In the Nature Genetics paper, we used phylogenetically corrected Fisher’s exact test to measure associations and account for population structure. For this method we inferred the phylogenetic tree of the viral sequence data, estimated ancestral viral sequences and compared them to the viral sequences at the tips of the tree. The presence and absence of changes was then tested for association with SNP rs12979860. However, in the current study we apply logistic regression and account for population structure by including viral and host genetic PCs as covariates in the analysis. This approach has recently been shown to be more powerful to detect association in genome-to-genome analysis (Correcting for Population Stratification Reduces False Positive and False Negative Results in Joint Analyses of Host and Pathogen Genomes; DOI=10.3389/fgene.2018.00266). Thus, focusing on a homogenous population (white ethnicity infected with gt3a) and using logistic regression has increased the power of our study to detect many more associations between the SNP rs12979860 and the HCV amino acids.

We have now expanded the second paragraph in the Discussion to compare the findings in this study with those in the Nature Genetics paper.

The value of including the Expanded Access Programme (EAP) subjects is unclear. EAP contributes only ~15% of the total subjects and differs from the BOSON group regarding important demographic and clinical characteristics (sex, prevalence of cirrhosis, HCV RNA levels), as well as genotype frequencies of rs12979860. If EAP subjects are retained in the revised paper, these differences should be considered in the analysis and addressed in the Discussion.

Although the EAP group represents a lesser proportion of the cohort, our aim has been to gather as many well-characterised samples in terms of viral sequence and clinical data as possible. As stated above, increasing the scale of the group under study enhances the confidence in any statistical analysis for identifying associations. Moreover, the viral sequence data was generated by similar methods in Oxford and Glasgow using target enrichment probe sets and next generation sequencing. The viral sequence data generated does not use HCV-specific primers and so the sequences produced are not biased. Putting together such a large number of sequences covering almost the entire length of the viral genome is not a trivial task. Additionally, we tested and found no association between the HCV phylogenetic tree (which estimates the population structure of the virus) and the host INFL4 SNP rs12979860 (using treeBreaker as indicated in the subsection “Host and virus genetic structures” and Figure 1—figure supplement 3B). Furthermore, we also performed all of the analysis in the BOSON cohort alone and found similar association results that are reported in the Materials and methods. Finally, we have performed an analysis testing for association between the cohort of origin of the patient and viral amino acid variants and found no significant associations (see below). Therefore, we believe that there are sound reasons for combining the BOSON and EAP cohorts.

The reviewers have many questions and concerns about the statistical methods and the analyses.Multiple testing: FDRs of either 5% or 20%, as well as Bonferroni correction are employed. Unless there are compelling reasons for these different approaches (which should be stated), a uniform approach to multiple testing adjustment should be used. The Abstract states rs12979860 genotype associated with 4% of viral amino acid sites across the HCV polyprotein. Based on the discussion (that finding does not appear in the results), this result is based on a 20% FDR, whereas findings presented in the Results (and the previous paper) are based on a 5% FDR. A 20% FDR seems very high and needs to be justified.

Firstly, we have modified the Abstract to remove the statement on association with 4% of viral amino acid sites across the viral polyprotein. In addition, we have modified the text to use 5% and 10% FDR thresholds throughout the manuscript.

Genomic inflation factor for IFNL4 rs12979860 and 500 SNPs with similar frequencies: The genome inflation factor (represented by λ) is used to examine assumptions re: cryptic relatedness when a large set of SNP markers are tested for association with a dichotomous trait in a GWAS (https://www.ncbi.nlm.nih.gov/pubmed/11315092 https://journals.plos.org/plosone/article?id=10.1371/journal.pone.0019416), but it is unclear what is being analyzed here – association of viral variants with a trait (yes or no host SNP)? Are the assumptions about the distribution of viral variants the same as for distribution of host germline variants, for which this approach was developed? This is not a standard approach to test for unaccounted population structure or other biases and the logic for doing this is unclear. Can the authors provide a reference to support use of this method?A λ value of 2.16 is extremely high and indicates cryptic relatedness, but in this case that statistic is impossible to interpret it because the approach is not described adequately.

We have modified the text to simplify this section. We have tested presence and absence of each viral variant (amino acid or codon) against host SNPs (the rs12979860 SNP was converted to binary variables to indicate CC vs. non-CC genotypes and the other 500 SNPs with similar frequencies were similarly converted to binary variables to indicate homozygous for major allele vs. other genotypes). In other words we have performed 501 viral GWASs and in each GWAS the trait of interest is a host SNP (SNP rs12979860 and 500 other SNPs with similar frequencies).

In this revised version of the manuscript, we decided to present the results differently and to remove the genomic inflation factor. We have observed 42 association signals with *IFNL4* SNP rs12979860 at a 5% FDR, whilst 491 of the 500 SNPs with a similar allele frequency showed no association signal and 9 showed only one association signal. If host-virus population co-structuring was contributing to our results, we would expect to observe similar patterns between HCV amino acid variants and the other host SNPs frequency-matched to *IFNL4* SNP. Similarly, only the *IFNL4* SNP rs12979860 showed a distribution of observed P-values different from the distribution of P-values for a null hypothesis of no associations, as shown on the qqplots. We moved the qqplot figure to a figure supplement and added a Manhattan plot of the associations (Figure 1).

To answer the reviewer’s question, we detail as follows our assumptions for the genomic inflation factor as presented in the earlier version. For each of the viral GWASs against the 500 frequency-matched host SNPs, we calculated a genomic inflation factor. Only the viral GWAS with SNP rs12979860 as the outcome variable had inflated P-values, while none of viral GWASs with any other host SNP (outside of the IFNLs loci) as the outcome variable had inflated P-values. If co-structuring of host and virus populations or cryptic relatedness or other systematic biases were driving the inflation in the P-values of viral GWAS against host rs12979860 SNP, we would expect to observe the same effect for other host SNPs. As Figure 1—figure supplement 6A show, our viral GWAS P-values against all other host SNPs follow the expected null hypothesis of no association.

Principal component analysis: What is used for PCA in the host? There is no explanation and the plot differs from those used for GWAS, where study samples are plotted in relation to reference populations.

We used FlashPCA to perform PCA on the host genotype data. We performed this analysis only in the studied cohorts (all with self-reported white ethnicity) and did not include reference populations. This was to ensure that we were capturing lower level population structures in our cohort that may otherwise be lost in the context of global population structure. It is a standard analysis to do in human genetics with a homogenous cohort, even if it might not be the most reported one in the literature.

Which viral PCs were included? Were the PCs used as continuous variables in any models?

Viral sequence data were converted into binary variables which was then used for PCA. We used the first two PCs (as continuous variables) in all our viral GWASs unless indicated otherwise in the text.

(Note: "principal" is spelled as "principle" in several places.)

We modified the text accordingly.

Assessment of Confounding, Interaction and Mediation: Assessment as to whether adjustment for potential confounders (e.g. sex, age, study [EAP or BOSON], cirrhosis) is needed. Were genotypes associated with any patient characteristics? E.g. age or cirrhosis status?Stratified analyses should be performed to identify possible interactions for those variables, especially sex (interaction between IFNL4 genotype and sex has been reported for associations with hepatic fibrosis).

We assessed the impact of the above potential confounders on the viral amino acid variation. We performed four separate viral GWASs using logistic regression (one for each possible confounder). Presence and absence of viral amino acids was used as the response variable and 2 viral PCs, 3 host PCs and *IFNL4* SNP rs12979860 (CC vs. non-CC genotype) were added as covariates. FDR was calculated for each viral GWAS independent of the others. At 10% FDR there were no significant associations between cirrhosis status, cohorts (BOSON vs. EAP), gender and age, and any of the viral amino acid variants.

We included the following sentence to the Results section: “To test for possible confounders we added separately the cirrhosis status, cohorts (BOSON vs. EAP), gender and age to the model as covariates. These covariates were not associated with any specific amino acids at 10% FDR (data not shown).”

In a separate viral GWAS, we investigated whether the interaction between *IFNL4* SNP rs12979860 genotypes and gender had an impact on viral amino acid variation. We used the same procedure as above, adding 2 viral PCs, 3 host PCs, *IFNL4* SNP rs12979860 (CC vs. non-CC genotype) and gender as covariates. At 10% FDR there was no significant association between any viral amino acids and the interaction term of *IFNL4* SNP rs12979860 and gender.

We also performed another analysis where we compared the impact of *IFNL4* SNP rs12979860 (CC vs. non-CC genotypes) on viral amino acid variation, using two different logistic regression models. In model one, we used 2 viral PCs and 3 host PCs as covariates and in the second model we also added age, gender, cirrhosis status and cohort indicator as covariates. The P-values for the association between *IFNL4* SNP rs12979860 and the viral amino acid variants are highly correlated between the two models as shown in Author response image 1.

Since the results were negative, we decided not to include the two former interaction analyses in the revised paper to keep it as simple as possible.

The association of IFNL4 genotype with the frequency of HCV polymorphisms could reflect an effect of IFNL4 on viral replication rates. To assess that possibility, the investigators should compare the results of two logistic regression models: one that does and one that does not include HCV RNA as an additional covariate to IFNL4. Otherwise these paired models should include identical adjustments.

We agree that *IFNL4* SNP rs12979860 is associated with viral load such that CC patients on average have higher viral load and this could indicate the impact of *IFNL4* on viral replication rates. Higher levels of replication in CC patients could lead to more within-patient diversity, but we have shown that non-synonymous codon changes are much more likely to be associated with *IFNL4* SNP rs12979860 compared to synonymous codon changes. Additionally, unless there is a form of selection acting on specific amino acids, one cannot explain consistent changes to the same amino acid across multiple patients with the same *IFNL4* genotype.

However, we performed the above comparison suggested by the reviewer. In the first model we included 2 viral PCs and 3 host PCs as covariates and in the second model we also added log10(viral load) as a covariate to the model. We plotted the P-values of associations between *IFNL4* SNP rs12979860 and the viral amino acids from the two models against each other. Adding log10(viral load) as a covariate reduced the P-values very slightly as shown in Author response image 2, but the impact on the analysis was minimal. At a 5% FDR 37 viral sites were associated with host INFL4 SNP, increasing to 68 at 10% FDR.

**Author response image 2. respfig2:** 

Other comments re: statistical analysesSubsection “IFNL4 SNP has a widespread impact on the viral amino acids” first paragraph: how was the "expected median" computed?

To estimate the genomic inflation factor we calculated the median of the observed chi-squared test statistics and divided it by the median of the chi-squared distribution with one degree of freedom which is 0.4549 (expected median). The P-values were converted to chi-squared test statistics using qchiseq(1-p, df=1) in R. However we removed this section from the current version to simplify the presentation of the Results as indicated above.

Subsection “IFNL4 SNP has a widespread impact on the viral amino acids” second paragraph: was does "frequency matched" mean? With the same MAF?

We modified the text as follows: we performed the same tests against HCV amino acids for 500 host SNPs from across the human genome with a minor allele frequency (MAF) similar to *IFNL4* SNP rs12979860 MAF, further referred to as the “500 frequency-matched SNPs”:

Subsection “IFNL4 SNP has a widespread impact on the viral amino acids” first paragraph”, third paragraph: please state outcome variable for the logistic regression models.

We have modified the text to indicate that the presence and absence of viral amino acids was used as the outcome variable in the logistic regression models.

Subsection “IFNL4 SNP has a widespread impact on the viral amino acids”, third paragraph: please define FDR and give reference.

We have modified the text as follows: at 5% false discovery rate (FDR) which indicates the expected number of false positives among discoveries (significant associations). We added a reference (Benjamini and Hochberg, 1995).

Subsection “Statistical analysis”, third paragraph: please state clearly that the SNP was the outcome for the logistic models.

In the last paragraph of the subsection “Human and viral population structure”, we did not use logistic regression. We used treeBreaker software (Ansari and Didelot, 2016). This software uses Bayesian methodology to measure association between a tree and a phenotype of interest. In brief, given a tree and tip phenotypes, it infers which clades on the tree, if any, have a distinct distribution of the tip phenotype from the rest of the tree.

Subsection “Statistical analysis”, eleventh paragraph: To obtain maximum likelihood estimates one needs to assume a normal distribution for log_10_(viral load) transformed data. IN my experience this assumption is not true for log_10_(viral load) transformed data. However, least squares estimates do not require this assumption.

The qqplot of the model residuals against standard normal distribution indicates that the assumption of normality is justified.

**Author response image 3. respfig3:** 

Subsection “Statistical analysis”, eleventh paragraph: when a line was fit through the log(OR) estimates, how were the standard deviations of the log(ORs) used? That uncertainty needs to be accommodated.

We have modified our analysis to account for the uncertainty associated with the log(ORs). To do this we used bootstrapping. We simulated 10,000 bootstrap datasets where the log(ORs) were simulated using a normal distribution with mean set to the log(ORs) and standard deviation set to the standard error of the estimate. For each dataset we fitted a linear regression and estimated the slope of the fit. The empirical bootstrap 95% confidence interval of the slope of the line was in (0.69,0.99) which does not include 1.

Other general comments:The presentation is hard to follow with most data presented as minimally annotated supplementary materials with limited legends provided separately. Providing more detailed legends next to corresponding figures and tables should make it easier to follow.

The *eLife* submission process requires us to submit supplementary figures and legends separately. We hope that in the final version of this manuscript, legends will be provided next to the corresponding figure or table.

In the Discussion, the limitations of the study should be explored.

We have included a paragraph (second last) in the Discussion on the limitations of the study.

[Editors' note: further revisions were requested prior to acceptance, as described below.]

The manuscript has been improved but there are some remaining issues that need to be addressed before acceptance, as outlined below:Overall, the authors were responsive to reviewer comments and the paper is much improved. The analytical approach remains complicated and the paper is still challenging to read. The authors should consider and address the following comments.Multiple testing: The authors eliminated use of the Bonferroni correction and a false discovery rate (FDR) of 20%. The paper still presents two different FDR thresholds (5% and 10%) for many analyses and the reason for doing so is unclear. It would be simpler to report a single set of results based on a 5% FDR, the threshold used in the previous publication from this group.

False discovery rate allows us to quantify the expected number of false positives in our results. Choosing a low FDR will ensure that there are very few false positives, but setting a low FDR value also risks reducing the number of true positives. In the present version of the manuscript, we decided to report both 5% and 10% FDR thresholds. The reason for doing so is that a large number of associated viral amino acids were required for analyses to be performed with sufficient power. This could only be achieved with a 10% FDR. In our previous publication, we also used two different FDR thresholds (5% and 20%). The lower 5% FDR is applied in this present work to help with comparisons to our previous report.

Normality of viral load data: It is not clear from a visual inspection of the Q-Q plot that these data are normally distributed. A P-value for fit would be a more objective measure.

This comment is related to the first set of reviews that we received. “Subsection “Statistical analysis”, eleventh paragraph: To obtain maximum likelihood estimates one needs to assume a normal distribution for log_10_(viral load) transformed data. IN my experience this assumption is not true for log_10_(viral load) transformed data. However, least squares estimates do not require this assumption.”

We answered this comment in the rebuttal by presenting a qq-plot. No modification was added to the manuscript. Here, we explain the reason why we are confident that the tests performed in the manuscript are valid.

In small samples most statistical methods do require distributional assumptions, and the case for distribution-free rank-based tests is relatively strong. However, in large data sets (which is the case for our study), most statistical methods rely on the Central Limit Theorem, which states that the average of a large number of independent random variables is approximately normally distributed around the true population mean. It is this normal distribution of an average that underlies the validity of the t-test and linear regression, but also of logistic regression and of most software for the Wilcoxon and other rank tests.

The t-test and linear regression compare the mean of an outcome variable for different subjects. While these are valid even in very small samples if the outcome variable is normally distributed, their major usefulness comes from the fact that in large samples they are valid for any distribution (Lumley et al. The importance of the normality assumption in large public health data sets. Annu Rev Public Health. 2002;23:151-69).

Second paragraph of the Introduction “substitutes a proline for a serine […]”: Terczyńska-Dyla et. al state, “an amino-acid substitution in the IFNλ4 protein changing a proline at position 70 to a serine (P70S) […]”. To this reader, that means a serine is substituted for a proline. Alternatively, the authors might use the language of Terczyńska-Dyla et. al to describe this variant.

We modified the text to read “the IFN-λ4 protein, which changes a proline residue at position 70 (P70) to a serine residue (S70)”. We are not using the P70S nomenclature elsewhere in the article so would prefer not to introduce it here. We hope the change will make the description of the variant clearer and accurate.

Subsection “Host and virus genetic structures”: Without any explanation, it is unclear how to interpret the Bayes factors of 249 and 1.1.

We added the following sentences: “see Materials and methodsfor an explanation on how to interpret Bayes factor’ and in the Materials and methods section: ‘Bayes factor is a summary of the evidence provided by the data in favour of one model compared to another. […] Bayes factor: 10 to 100, strong evidence against null model and in favour of the alternative model. Bayes factor: >100, decisive evidence against null model and in favour of the alternative model.”

How the patients are divided into IFNL4-null, S70 and P70 groupings could be clearer. Supplementary file 7 would present that information if the groups were arranged together and labeled.

We modified the Supplementary file 7 to include a row labeling the different haplotypes into the grouping that we made. Each haplotype is now linked to the protein predicted to be produced.

We also added the following sentence in the Materials and methods section: “HCV-infected patients were classified into three groups according to their predicted ability to produce IFN-λ4 protein: (i) no IFN-λ4 (two allelic copies of IFN-λ4-Null, N_BOSON_=145, N_EAP_=41), (ii) IFN-λ4–S70 (two copies of IFN-λ4-S70 or one copy of IFN-λ4-S70 and one copy of IFN-λ4-Null, N_BOSON_=48, N_EAP_=7), and (iii) IFN-λ4-P70 (at least one copy of IFN-λ4-P70, N_BOSON_=218, N_EAP_=26).”

Subsection “Host and virus genetic structures”, ninth paragraph – The co-submission by Chaturvedi et al. has been accepted for publication and might be referenced here.

We added the following sentence: (also see ‘Adaptation of hepatitis C virus to interferon lambda polymorphism across multiple viral genotypes’ by Chaturvedi *et al.* in this issue).